# Plasma generated ozone and reactive oxygen species for point of use PPE decontamination system

**Min Huang**[1], **Md Kamrul Hasan**[1], **Kavita Rathore**[1], **Md Abdullah Hil Baky**[1], **John Lassalle**[1], **Jamie Kraus**[1], **Matthew Burnette**[1], **Christopher Campbell**[1], **Kunpeng Wang**[2], **Howard Jemison**[2], **Suresh Pillai**[3], **Matt Pharr**[1], **David Staack**[1] *

1 Department of Mechanical Engineering, Texas A&M University, College Station, Texas, United States of America, 2 LTEOIL LLC, Houston, Texas, United States of America, 3 Center for Electron Beam Food Research, Texas A&M University, College Station, Texas, United States of America

* dstaack@tamu.edu

**Data Availability Statement:** All relevant data are available on OAKTrust: https://oaktrust.library.tamu.edu/handle/1969.1/195448.

**Funding:** The author(s) received no specific funding for this work.

## Abstract

This paper reports a plasma reactive oxygen species (ROS) method for decontamination of PPE (N95 respirators and gowns) using a surface DBD source to meet the increased need of PPE due to the COVID-19 pandemic. A system is presented consisting of a mobile trailer (35 m3) along with several Dielectric barrier discharge sources installed for generating a plasma ROS level to achieve viral decontamination. The plasma ROS treated respirators were evaluated at the CDC NPPTL, and additional PPE specimens and material functionality testing were performed at Texas A&M. The effects of decontamination on the performance of respirators were tested using a modified version of the NIOSH Standard Test Procedure TEB-APR-STP-0059 to determine particulate filtration efficiency. The treated Prestige Ameritech and BYD brand N95 respirators show filtration efficiencies greater than 95% and maintain their integrity. The overall mechanical and functionality tests for plasma ROS treated PPE show no significant variations.

## 1. Introduction

There has been a global shortage of Personal Protective Equipment (PPE) due to the highly contagious nature of COVID-19 [1–4]. Surface DBD [5–7] techniques have been widely investigated in the field of decontamination and bioburden reduction for Personal Protective Equipment (PPE). Such technologies generate a partially or fully ionized gas, UV radiation, oxygen species (O, O3, and O2*), or oxygen-containing radicals (e.g., OH* and NO*), which lead to the inactivation process of microorganism [8, 9]. Kramer et al. [10] developed a portable DBD plasma system that is capable of reducing microbial loads on surfaces with a 300-liter treatment chamber. Moisan et al. [11] reported that the highest sporicidal effect can be achieved when spores are directly exposed to the plasma discharge. UV photons and highly reactive short-lived species (including accelerated ions and electrons, as well as uncharged particles such as excited atoms, molecules, and radicals) all participate in various inactivation

**Competing interests:** The authors have declared that no competing interests exist.

mechanisms [12]. The DBD reactor is commonly used to generate ozone in any form (either aqueous or gaseous ozone and on hard or porous surfaces). Ozone is a well-known powerful oxidizer, which can kill microorganisms effectively [13]. The goal of this work is to develop a system capable of using plasma-generated ROS to decontaminate PPE in a hospital-type setting. For this reason, a mobile system is preferred.

In this paper, we have reviewed the relevant literature to determine an appropriate dose of plasma ROS, namely ozone, for sterilization. Ozone has been recognized as an effective biocide, having several attractive features due to its antiviral profile, (relatively) short half-life, and gaseous and diffusive nature. The paper includes detailed information about the ozone generation technique, as well as systematic processes of plasma ROS exposure to PPE/materials in a glovebox and a trailer. Further, this paper discusses standard methods for functionality material testing and performance of treated material samples. The objective of this paper is to provide a comprehensive assessment of the plasma ROS-based decontamination of PPE/material using a preexisting mobile trailer system. Analysis of results from standardized filtration and fit/strap testing indicate that the integrity of the respirators is maintained over decontamination cycles.

## 2. Literature review

An extensive literature review was undertaken to examine the amount of ROS needed for sterilization. Ozone can be used as an oxidant and disinfecting agent. It has key benefits such as oxidization of odors [14], control of airborne micro particulates [15], sterilization of microorganisms [16–20], increment of food shelf life [21, 22], and improvement in the growth rate of plants, flowers, poultry, etc [23]. The ozone dose is defined in this work as the product of ozone concentration on the microorganism (C) and the contact time (t). The results of microorganism survival fraction (SF) versus ozone dosage were obtained from the exponential decay model. The Survival Fraction (SF) is defined by [20]:

$$SF = \frac{N_s}{N_0} = e^{-KCt}$$

where, $N_s$ = Concentration of surface viruses survived after exposure to ozone (PFUs/mL), $N_0$ = Concentration of surface virus before exposure to ozone (PFUs/mL), C = Ozone concentration (ppm), t = Ozone contact time (min), and K = Virus susceptibility factor (1/ppm-min).

In previous studies, ozone has been found effective against specific diseases and pathogen like Hepatitis A; Enteroviruses; HIV; MS2 Coliphage; SA11; Poliovirus Type 1 and Type 3; Rotaviruses; Influenza viruses; enteric viruses; Rhinoviruses; and the Norwalk virus (NV) [24]. The NV can be inactivated in a closed environment of ozone with 20–35 ppm concentration in 30–70 minutes. In addition, it was found that the virus could be inactivated more efficiently in a high humidity (>80%) environment compared to an ambient humidity of 45–50%. Coronaviruses have abundant cysteine in their spike proteins that may be easily and safely exploited with ozone (or other oxidation) therapy [25]. The study has shown that ozone at a concentration of over 100 ppm in a high humidity environment was highly virucidal [13] against four RNA viruses namely (1) HVJ, (2) Theiler's murine encephalomyelitis virus (TMEV), which is a coronavirus, (3) Reo type 3 virus (RV), and (4) murine hepatitis virus (MHV). A zero level of infectivity was obtained for the HVJ and TMEV virus samples within one to three hours if treated with 200 ppm of ozone at 80% humidity. According to Occupational Safety and Health Administration (OSHA), in case of light work, air standards of 0.1 ppm for 8 hours per day (or 0.2 ppm for no more than 2 hrs.) are the safe limits for ozone exposure [26].

**Table 1. A summary of O₃ concentrations for different strain log reduction.**

| Ref. | O₃ (ppm) | RH% | Exposure time (min) | O₃ dose (ppm-min) | Strain(s) | K (susceptibility factor) (1/ppm-min) | Effectiveness (log reduction) |
|---|---|---|---|---|---|---|---|
| [20] | 1.2 | 85 | 20 | 24 | Φ6 (with envelope lipid, ATCC 21781-B1) | 0.19 | -2 |
| [20] | 1.2 | 85 | 23 | 27.6 | ΦX174, ATCC 13706-B1 | 0.16 | -2 |
| [20] | 1.2 | 85 | 44 | 52.8 | MS2 [ATCC] 15597-B1 | 0.08 | -2 |
| [20] | 1.3 | 85 | 81 | 105.3 | T7 ATCC 11303-B | 0.04 | -2 |
| [33] | 631 | 20 | 0.25 | 157.75 | Escherichia coli (EC) | 0.03 | -2.1 |
| [33] | 1500 | 20 | 0.25 | 375 | Staphylococcus aureu (SA) | 0.01 | -2.5 |
| [34] | 20 | 70 | 20 | 400 | Poliovirus (PV) | 0.01 | -2.8 |
| [34] | 20 | 70 | 20 | 400 | FCV | 0.01 | -2.9 |
| [19] | 6.2 | 90 | 60 | 372 | Penicillium glabrum (PG) | 0.01 | -3 |
| [19] | 7.3 | 90 | 60 | 438 | Streptomyces (SM) | 0.01 | -3 |
| [19] | 9 | 90 | 60 | 540 | Penicillium chrysoenum (PC) | 0.01 | -3 |
| [19] | 9.9 | 90 | 60 | 594 | Rhodotorula glutinis (RG) | 0.01 | -3 |
| [35] | 25 | 70 | 20 | 500 | (Feline calicivirus) FCV | 0.01 | -3 |
| [35] | 25 | 70 | 20 | 500 | (Norovirus) NV | 0.01 | -3 |
| [36] | 25 | 90 | 20 | 500 | Methicillin-resistant Staphylococcus aureus (MRSA) | 0.01 | -3.1 |
| [36] | 25 | 90 | 20 | 500 | Bacillus cereus (BC) | 0.01 | -3.1 |

Table 1 shows the literature summary of O₃ effectiveness (log reduction) on strains with ozone dose (ppm-min). It is noted from Table 1 that there is no direct relationship between ozone dose and inactivation of different strains. However, a few trends can be inferred from the available data. The literature suggests that exposure time for gaseous ozone applications ranges from 1–85 minutes and concentrations of ozone ranges from 24–500 ppm-min to attain significant microorganism inactivation. The Virus susceptibility factor (K) is the largest for humidity in the range of 70–90 RH %.

Fig 1 shows the trend of average literature data for the effectiveness of strains with exposure to ozone dose (ppm-min). This suggests that bioburden reduction ($\geq$ 3-log) of PPE can be achievable with 500 ppm-min of ozone dose at a higher RH (%) condition (70–90%). This definition for bioburden reduction comes from FDA guidance documentation, stating that a proposed bioburden reduction system (Tier III systems) should be capable of achieving $\geq$ 3-log reduction in the case of non-enveloped viruses or vegetative bacteria [27]. Therefore, 500 ppm-min per cycle as an effective bioburden reduction dose is a reasonable chosen dose set in our experiments. We expect that a longer dose of 1500 ppm-min (or equivalently 3 cycles) will be sufficient to achieve decontamination, defined as achieving $\geq$ 6-log reduction of microbial load. The averaged literature data drew in Fig 1 indicates there is sufficient support for the Tier III system. For Tier I and Tier II systems, however, it is necessary to perform direct measurement of decontamination.

While SAR-CoV-2 (or any corona virus) is not specifically addressed by this literature review viruses, similar enveloped viruses presented in Table 1 is Φ6. Vaccinia virus, vesicular stomatitis virus, yellow fever virus, and Sendai virus are all enveloped viruses and potential explanations for their ozone inactivation are the protein capsid damage and genome degradation. There is a great agreement on the formation of alterations, which are induced by ozone in the lipids and proteins present in the membrane of these viruses [28]. As a preferred method used for generation of ozone, corona discharge also provides another potential mechanism for virus decontamination. Although reactive oxygen and/or nitrogen species (RONS) play the most important role in sterilizing microorganisms, ultraviolet radiation induces direct damage

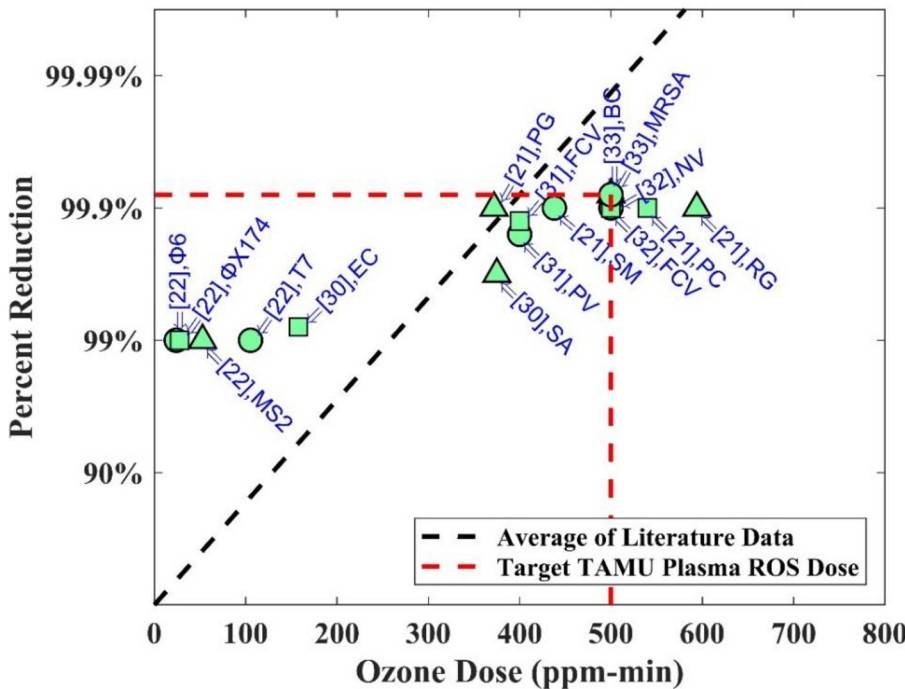

**Fig 1. Graph of percent reduction (%) of strains and ozone dose (ppm-min).**

to microorganisms and bacteria via breakage of DNA/RNA and chemical alterations of bases due to the absorption of high energy. More detailed information is covered in the section 3.1.2 regarding direct ozone induced and indirect ozone-generated oxidant species induced damages. Increasingly attention on ozone inactivation of enveloped virus is aware, and the mechanism has been discussed in recent researches [29–31]. Research also indicates that induced changes in the viral RNA genome, appearing as aggregation and fusion of influenza viruses (also known as enveloped viruses) were observed via SEM after N2 gas plasma treatment [32].

## 3. Experimental Setup

### 3.1. Plasma ROS generation method

**3.1.1. DBD plasma reactor.** The Aerisa devices (models 5350/5550) generate dielectric barrier discharges at atmospheric pressure conditions [6, 37]. Fig 2 shows the photograph of an Aerisa model 5350 with different view angles and a zoom-in image of the DBD surface. This device can be divided into two parts: the upper DBD reactor and the lower circuit box (Fig 2a and 2b). In the upper DBD reactor, there are five DBD tubes (tubes needed be changed every 3 months due to dust accumulation.), each consisting of an inner cylindrical perforated electrode (2 mm diameter), a middle quartz tube, and an outer meshed electrode. The lower circuit box converts the input voltage via a high-voltage transformer from 110 V to 3.2 kV for DBD generation. The DBD operates at a frequency of 60 Hz. The circuit box also contains a 5-step knob for selection of the winding ratio in the transformer and power selection settings. A detailed surface feature image of the electrode layout can be seen in Fig 2c. In the top mesh electrode, a square wire mesh is used with a mesh surface area of $1.5*1.5$ mm$^2$. The lengths of the electrode tubes are 0.28 m (Aerisa 5350) and 0.44 m (Aerisa 5550). The waveforms of applied voltage and discharge current for Aerisa 5350 are given in Fig 2d.

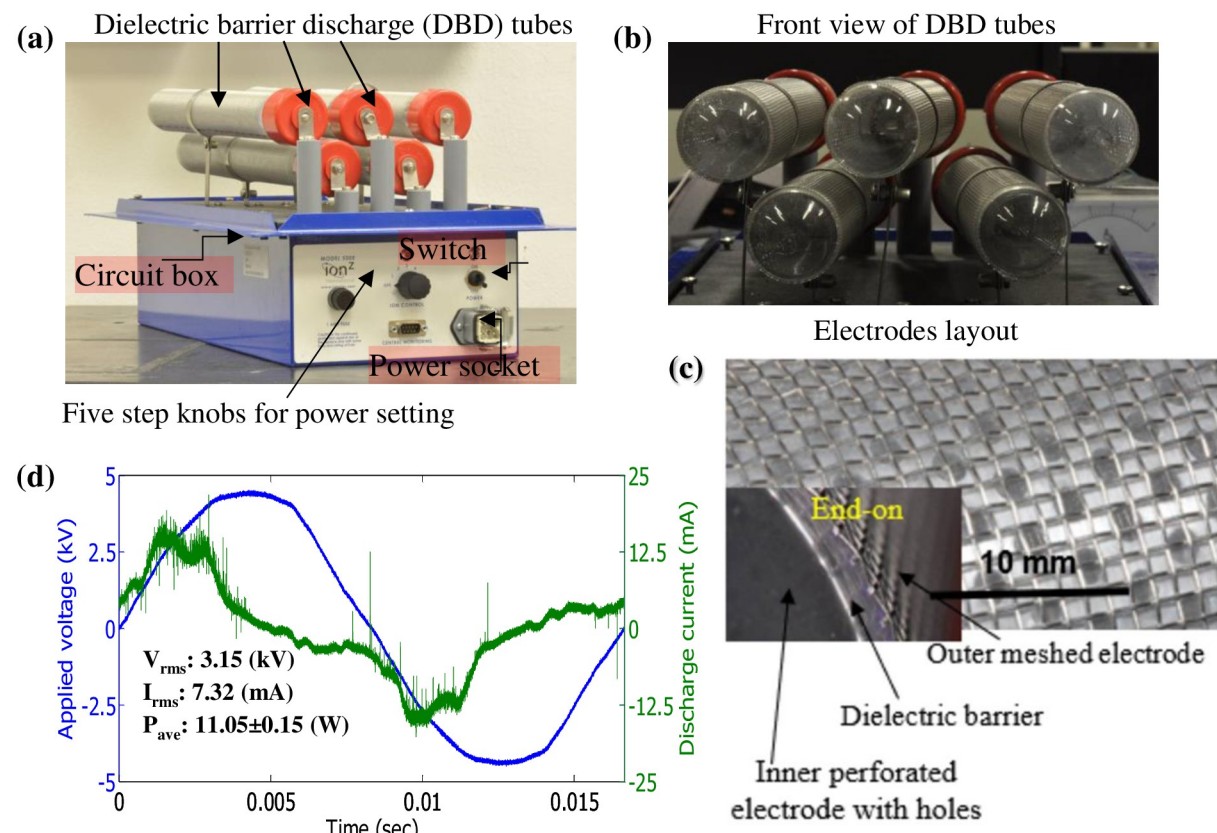

**Fig 2. Device configuration of Aerisa 5350/5550: (a) DBD setup, (b) front view of DBD tubes, (c) electrode surface, and (d) voltage-current profile.**

After the high voltage is applied on the Aerisa device, a faint light emitting from the resulting DBD plasma can be observed surrounding the electrode (Fig 3). This plasma is brightest at the maximum power knob setting (Fig 3a). Fig 3b shows the front view of the Aerisa device during operation. Many microdischarges can be observed in the zoomed image as shown in Fig 3c. The observed plasma is categorized as a filamentary DBD, due to the electrode-dielectric-electrode structure of the device and the occurrence of micro-discharges. These microdischarges tend to occupy a larger area at higher power. It should be noted that plasma is not generated over the entire surface of the DBD due to inconsistent gap distances between the meshed electrode and the dielectric quartz tube. This type of plasma generator is a non-proprietary design available from other manufacturers and an equivalent is the AtmosAir 508FC.

Line power ($P_{line}$) was measured directly using an electrical power meter and was in the range of 6–30 W for a single device. For the measurement of discharge power, a Lecroy (1000:1) 100 MHz, a high-voltage probe was connected to the outer electrode to measure the applied voltage. The transformer terminal was connected to the inner electrode, and it was modified to be electrically grounded. A 1.5 kΩ resistor was placed between the inner electrode and ground for measurement of the discharge current. Both the applied voltage and the current discharge were monitored and recorded by a Lecroy 204MXi oscilloscope.

The discharge current generated by Aerisa device was in the range of 5–20 mA. The measured line powers and discharge powers for the Aerisa device (Model No. 5530/5550) are listed

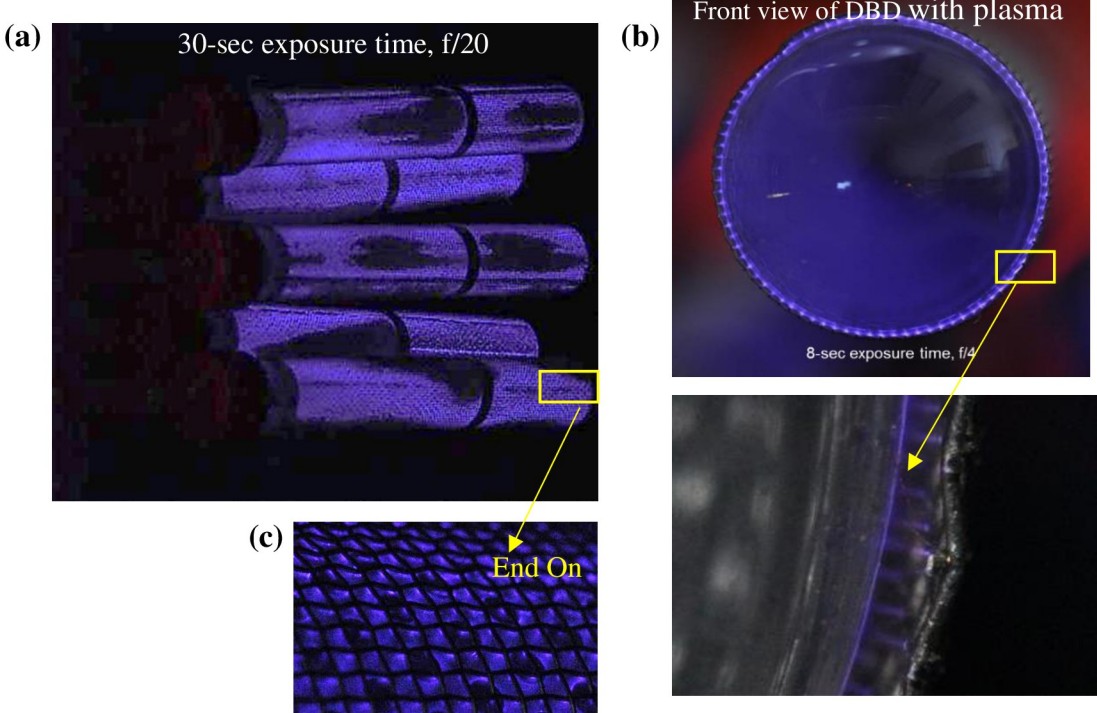

**Fig 3. Device configuration of Aerisa 5350: (a) full view of DBD plasma generated on the outer surface of tubes, (b) filamentary discharge at the front side of a DBD tube, and (c) surface microdischarge.**

in Table 2. The corresponding electrical efficiency at each power setting was also calculated and given in Table 2.

During operation the Aerisa devices generate various plasma and chemical species. Long lived species which are observable outside of the discharge zone are positive ions, negative ions and ozone. The ozone production rate is dependent upon discharge power and ranges from $1\times10^{20}$ #/s to $3\times10^{20}$ #/s. The positive ion and negative ion generation rates range from $2.7\times10^{10}$ #/s to $4\times10^{10}$ #/s and $4.2\times10^{10}$ #/s to $5.4\times10^{10}$ #/s respectively, as shown in Table 3. These rates were measured by placing a single generator in a controlled continuous flow system and measuring the downstream concentrations using either a ozone detector or biased faraday probe. Over time in a closed air system like used in the PPE decontamination systems

**Table 2. Measured line powers, discharge powers, and their corresponding electrical efficiency of Aerisa DBD generators.**

| Aerisa Model No. 5350 | | | | | |
|---|---|---|---|---|---|
| Power Setting | 1 | 2 | 3 | 4 | 5 |
| Line Power, $P_{line}$ (W) | 6.1±0.8 | 8.8±1.6 | 11.4±1.4 | 14.3±1.6 | 17.8±3.0 |
| Discharge Power, $P_{dis}$ (W) | 2.6±0.3 | 4.5±0.8 | 6.6±0.8 | 8.8±1.0 | 11.5±2.0 |
| Efficiency (%) | 42.6 | 51.1 | 57.9 | 61.5 | 64.6 |
| Aerisa Model No. 5550 | | | | | |
| Power Setting | 1 | 2 | 3 | 4 | 5 |
| Line Power, $P_{line}$ (W) | 10.3±0.9 | 14.5±0.8 | 19.1±1.4 | 24.3±1.3 | 30.5±2.6 |
| Discharge Power, $P_{dis}$ (W) | 6.8±0.9 | 10.5±0.8 | 14.1±1.0 | 18.3±1.0 | 23.2±2.0 |
| Efficiency (%) | 66.0 | 72.4 | 74.8 | 75.5 | 76.1 |

**Table 3. Downstream ion concentration in 620 cfm (0.3 m³/s) flow ambient air (24.5°C, 40%).**

| 5350 Power Knob Setting | Ozone Concentration (#/mL) | Negative Ion Concentration (#/mL) | Positive Ion Concentration (#/mL) | Ozone Flow Rate (#/s) | Negative Ion Flow Rate (#/s) | Positive Ion Flow Rate (#/s) |
|---|---|---|---|---|---|---|
| 1 | $3.29 \times 10^{14}$ | 140,000 | 89,000 | $1 \times 10^{20}$ | $4.2 \times 10^{10}$ | $2.7 \times 10^{10}$ |
| 5 | $1.05 \times 10^{15}$ | 180,000 | 130,000 | $3 \times 10^{20}$ | $5.4 \times 10^{10}$ | $4 \times 10^{10}$ |

the overall concentrations will increase over time, there will be degradation of the species and the production rates will be slightly affected by the higher initial species concentrations. Both ozone, other reactive species, and charged species might contribute to bioburden reduction. However, significantly more literature is available on the efficacy of ozone in decontamination. For this reason, in our bioburden reduction testing ozone concentration and exposure time, described as ppm-min dose, was used as the metric for determining a cleaning cycle.

**3.1.2. Reaction kinetics.** The microdischarges are generated by excitation and dissociation processes triggered by energetic electrons, electron multiplication, space charge accumulation processes, and ionization. The pulse energy of the microdischarge has an essential influence on the chemical reactions leading to ozone generation efficiency [38]. The reaction kinetics of ozone generation was modeled by Kogelschatz [39]; note that the dissociation rate coefficient of $O_2$ by electron impact depends on the energy distribution of the electrons in the discharge. This coefficient is usually treated as a function of the reduced electric field, which is defined as the ratio of electric field strength per unit gas density (E/n). The optimum reduced electric field for ozone formation from air is about 200 Td [38]. The initial step in the formation of $O_3$ is therefore the electron impact dissociation of molecular $O_2$ [40]

$$e^-(3\ eV) + O_2 \rightarrow O_2^* + e^- \tag{1}$$

$$O_2^* \rightarrow O + O \tag{2}$$

$$O + O_2 \rightarrow O_3 \tag{3}$$

$$e^-(0.3\ eV) + O_2 \rightarrow O_2^- \tag{4}$$

Negative ion production via electron attachment to $O_2$ dominates at lower electron energies, relative to electron-induced dissociation reactions. Peyrous [41] studied the effect of RH on ozone production in air. It was concluded that when water vapor is present, there is a probability of production of $H_2O_2$, $HNO_2$, and $HNO_3$ in air. The presence of water vapor and increasing temperature reduces ozone production.

Although the ozone concentration is reduced by 60% with an increase of R.H. from 0 to 10% [41], ozone concentration can effectively reach a high level with compensation of increasing time and DBD generator. Meanwhile, with high humidity present, increased reactive nitrogen species (RNS) also have been reported, though the hydrogen peroxide ($H_2O_2$) is still the main gaseous product above the ozone and the two acids ($HNO_2$, $HNO_3$) [41]. It is also known that the hydrogen peroxide resulted from recombination of two OH molecules contribute to the oxidizing reactions in biological cells, leading to the inactivation of virus. Even nitrites ($NO_2^-$) and nitrates ($NO_3^-$) increased with high humidity, the major plasma generated species responsible for inactivation is still ROS, with ions and RNSs having a secondary complementary role [42]. The positive correlation of ion density and bacterial inactivation has also been investigated [42]. Molecular $O_3$ reacts with organic compounds indirectly, generating high reactivity of the free radicals ($OH^*$ and $HO_2^*$), and then inactivate organisms. Hydroxyl

radicals ($OH^*$) are formed in discharges ignited in humid atmosphere with any ion with ionization energy above the $H_2O$ ionization threshold. $OH^*$, along with other free radicals ($HO_2^*$, $O_2^{*-}$, $H_2O_2$), then react with organic compouds [43], as seen in Eqs 5–7:

$$OH + RH \rightarrow R \cdot + H_2O \tag{5}$$

$$R \cdot + RH \rightarrow RO_2 \tag{6}$$

$$RO_2 + RH \rightarrow RO_2H + R \tag{7}$$

With the presence of nearby organics, hydroxy radicals react with and result in chain oxidation and thus damage of cellular membranes and other cell components [43]. While indirect reaction responsible for inactivation supported by some authors [44], some other literature claimed direct reaction with organic compounds dominates in the inactivation process [45]. Direct reaction assumed that $O_3$ reacts with a double bond to form initially an unstable "molozonide" (II), which rearranges to the more stable ozonide (III), an example is shown in Eq 8 [46].

Two general equations representing direct and indirect reaction with organic compounds M are shown below as summary, $M_{OX}$ stands for oxidized compounds:

$O_3$ reacts directly with organic compounds M (e.g. double bonds):

$$O_3 + M \rightarrow M_{OX} \tag{9}$$

$O_3$ indirectly reacts with organic compounds M

$$O_3 \xrightarrow{H_2O} OH^* \xrightarrow{M} M_{OX} \tag{10}$$

Mahfoudh et al [47]. showed that under dry gaseous ozone exposure, $O_3$ molecules could efficiently inactivate certain types of spores (G. stearothermophilus) compared to others such as B. atrophaeus. The differences in the inactivation rate depend presumably on the nature or arrangement of their constituents, essentially the chemical composition of their coats (and inner membrane). Another group of authors published that the main inactivation species such as $H_2O_2$ and possibly free hydroxyl radicals are able to access the spore core and DNA parts, also spore swelling under humidified media [48, 49].

### 3.2. Glovebox and trailer setups

For small scale testing, a metal glove box (Fig 4) was used as a reaction volume to treat the PPE samples. It is a powder coated steel box (0.68 m x 0.51 m x 0.51 m) with a slanted window to avoid optical distortion. The glove box has a transfer window (0.21 m x 0.21 m) which is used for releasing the ozone gas after the experiments. The windows are made of clear acrylic and sealed with rubber (Ethylene Propylene Diene Monomer) gaskets. The volume of the chamber

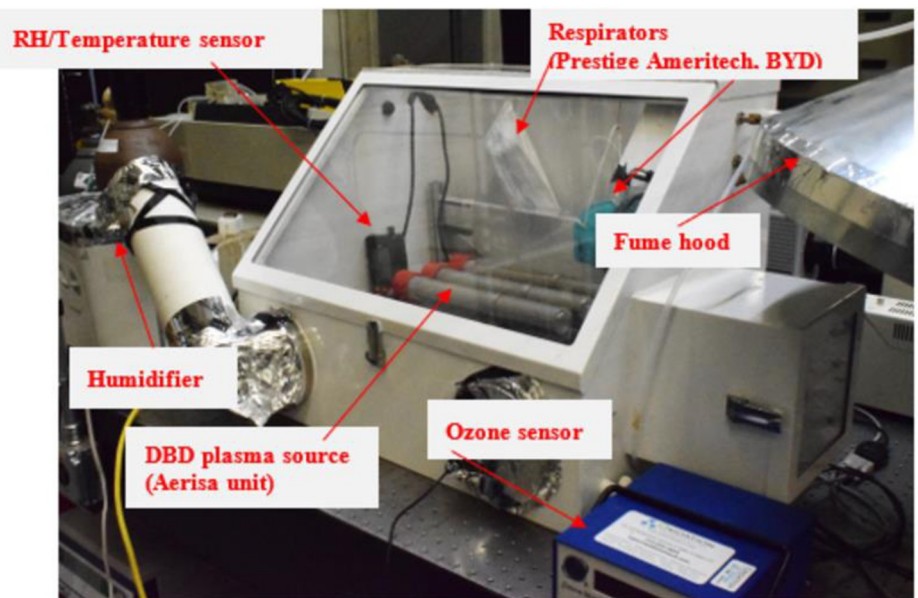

**Fig 4. Schematic view of glove box testing setup.**

is 0.15 m$^3$. The ozone-incompatible latex gloves from the glove box ports were replaced with aluminum foil, and one of the two ports was used to connect a humidifier to control the humidity inside the box.

The glove box is equipped with two ball valves, allowing gas to flow in and out of the chamber. An ozone detector (O$_3$ Ozone Monitor; 106-L by 2B Technologies) was connected through one of the valves using polyurethane tubing with a 1/4-inch diameter, permitting real-time data monitoring and logging. The gas flow rate to the detector is around 1000 mL/min. A small electric fan was used for better air circulation inside the chamber. The sensor for the temperature/ relative humidity (RH) was positioned in the glovebox, and a Honeywell HCM-750 humidifier was used to control the RH values. The temperature was monitored but not controlled directly. A fume hood was installed close to the small window for ozone gas ventilation to protect the researchers. Initial tests were performed to expose the ozone dose onto different types of N95 respirators (BYD N95 and Prestige Ameritech) with a low dose level of 500 ppm-min (1 decontamination cycle) and a high dose level of 1500 ppm-min (3 decontamination cycles). Fig 5 represents a typical ozone concentration variation with time for one and three cycles of decontamination. The ozone generator was operated at full power in order to achieve the desired level of the ozone dose at 75% RH. In addition, other PPE materials, straps of respirators, and two types of respirators, namely 3M 9502+ and 3M 8200, were subjected to ozone exposure of 200, 400, 800, and 1600 ppm-min at a maximum ozone concentration of approximately 20 ppm. The objective was to evaluate the effect of different ozone dose exposure on PPE materials and strap specimens. After the experiments, an ozone concentration of < 0.1 ppm was ensured before handling the materials inside the glovebox for personnel safety according to OSHA guidelines.

To determine the effectiveness of ozone for viral decontamination levels, a mobile trailer was modified to facilitate larger-scale tests (see Fig 6). The trailer (35 m$^3$ volume) is a bumper-pull Class IV hitch trailer with a 6350 kg gross vehicle weight rating (GVWR). The container dimensions are 2.59 m wide, 2.4 m tall, and 6.09 m long, and it has two external fans driven by non-sparking 10-amp motors (Dayton 43Y138) providing 2.69 m$^3$/min of ventilation and 0.61

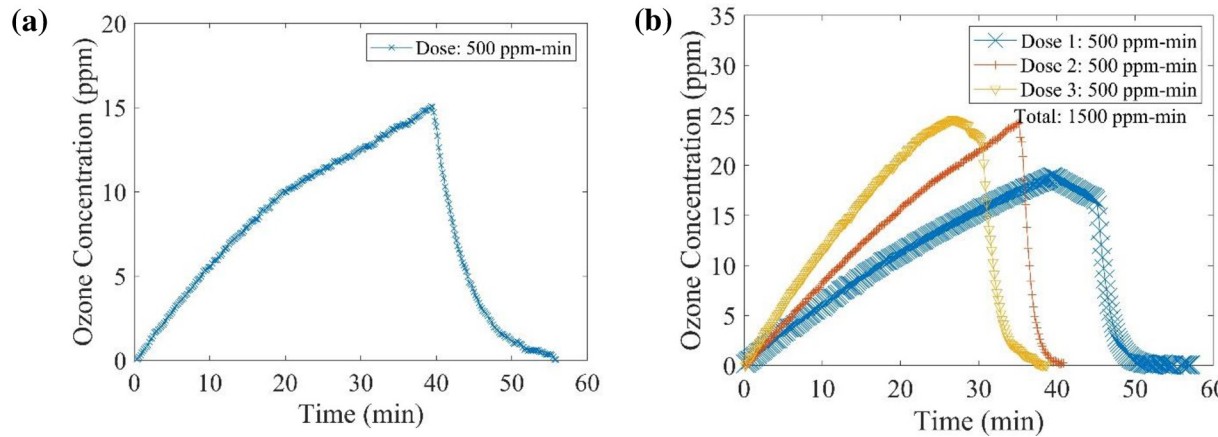

**Fig 5. Ozone concentration versus time with delivered doses of (a) 500 ppm-min and (b) 1500 ppm-min during N95 respirators decontamination (Glovebox).**

m/s face velocity. It has a Gasoline Generac (GP17500E) portable generator that is capable of providing power for remote operation of trailer lighting and ventilation systems and other electronic lab devices. The trailer has locks on both the rear doors and on the receiver hitch to prevent tampering or theft.

Testing was conducted to expose different types of PPE to reactive oxygen species. Ozone dose was used as an index of treatment intensity, with a low target dose level of 500 ppm-min (1 decontamination cycle) and a high dose level of 1500 ppm-min (3 decontamination cycles). Several surgical masks, two different types of N95 respirators, and personal protective gowns were arranged (gap of 2.54 cm) uniformly on the metal wire shelves with a polyurethane plastic coating (ClosetMaid™) in the trailer as shown in Fig 6b. Six Aerisa DBD generators (five Aerisa 5350 units and one 5550 unit) were mounted on rails near the roof of the trailer. A total of six of each type of respirator and the gowns were treated: three at 500 ppm-min and three at 1500 ppm-min. Treatments were conducted in multiple rounds of approximately 500 ppm-min each over several days. Humidity and temperature were monitored throughout the experiments in order to maintain low relative humidity variations ($\Delta RH < 20\%$) and temperature variations ($\Delta T < 10$ °F). Relative humidity and temperature varied from 75% to 95% and 24°C to 29°C (75°F to 85°F) respectively during treatment due to changes in the ambient environment. Commercial box fans were placed in the trailer at one end of the PPE shelves to promote uniform distribution and mixing of ozone. After closing the doors and sealing the grates over the exhaust fans, the DBD generators were turned on remotely from outside to begin the ozone generation process. After turning off the ozone generator, the ozone concentration began to drop due to slight leaks in the trailer and self-decomposition to oxygen. Fig 7 shows the changes in ozone concentration (a) and temperature and humidity (b) during a typical round of treatment. The humidity was utilized to increase humidity with time. The operator was not allowed to be present in the trailer during the experiment due to the hazardous ozone concentration. Each ozone generator produced around 0.26 L/hr of ozone on average in the trailer.

## 3.3. Materials and material testing method

**3.3.1. Material samples.** Ozone exposure experiments for material testing were performed on N95 Respirators (3M 8200, Prestige Ameritech, and BYD), KN 95 respirators (3M

(a)

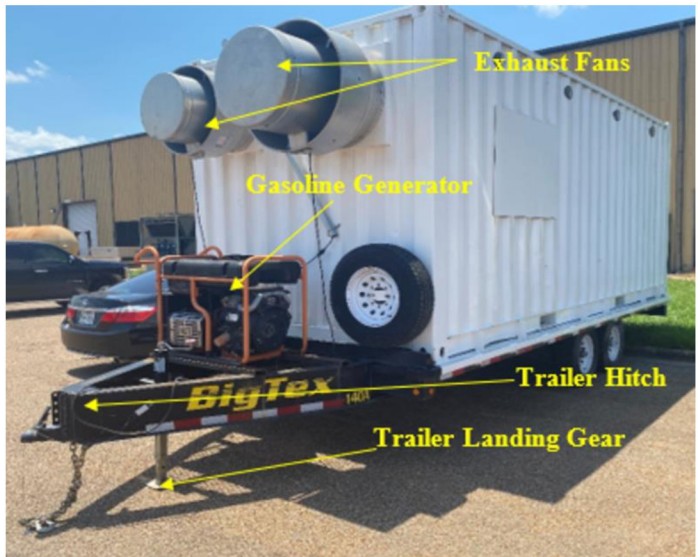

(b)

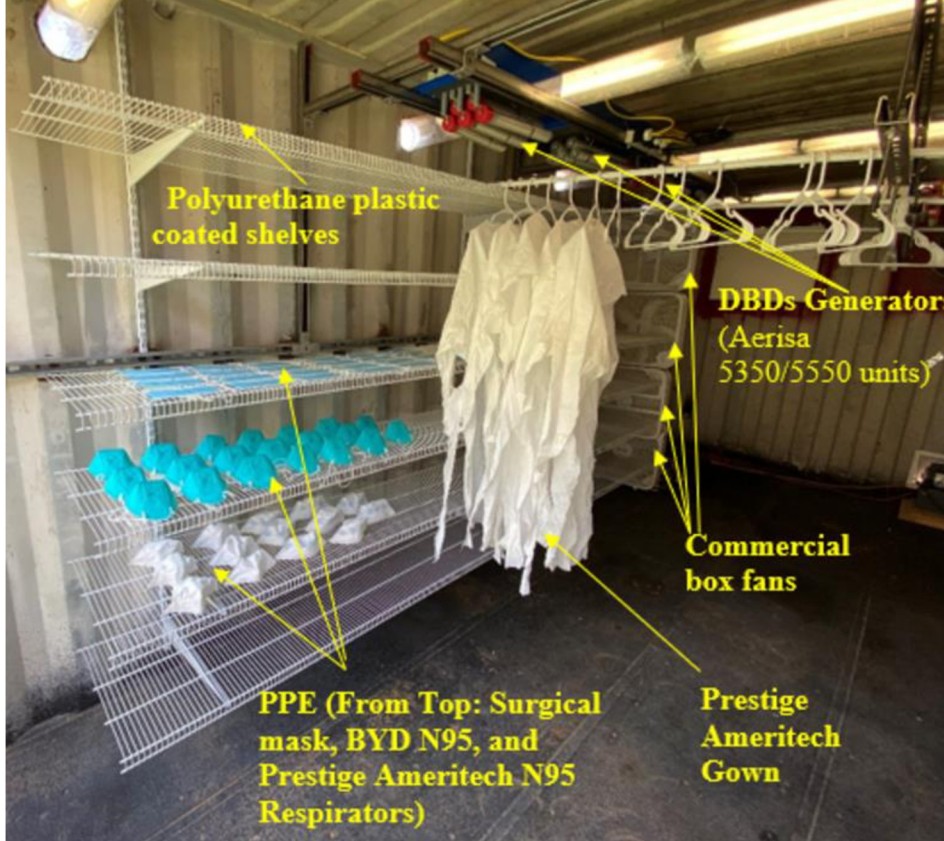

**Fig 6. (a) Outside view of the trailer and (b) Inside view of the trailer with PPE (surgical masks, respirators, and gown) arrangement.**

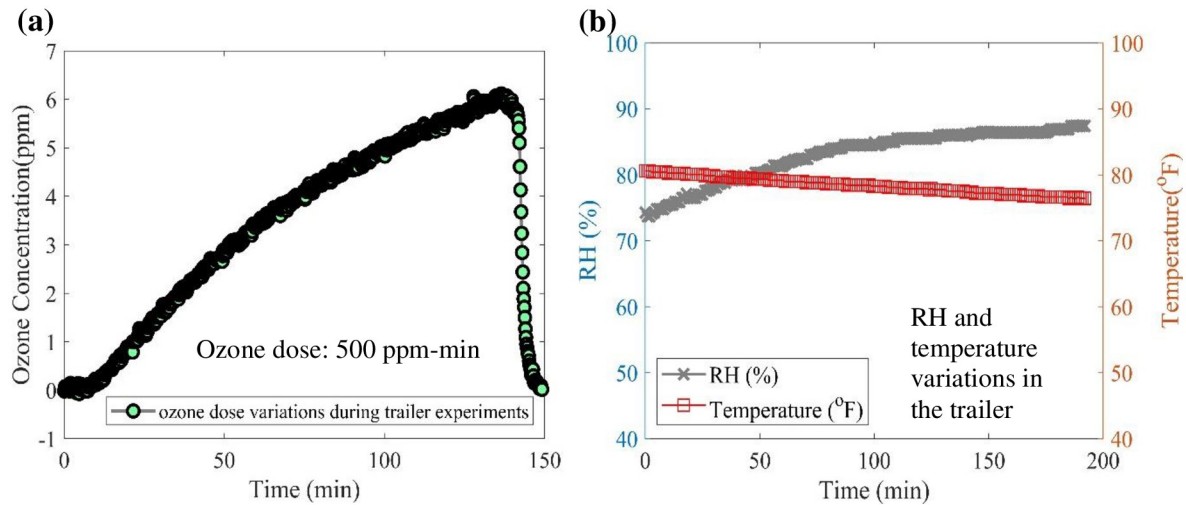

**Fig 7. (a) Ozone concentration variations with time during PPE exposure and (b) RH and temperature variations (trailer).**

9502+), gowns (Proxima and Prestige Ameritech), and raw materials of polypropylene and polyester. An initial set of baseline control samples were stored with no ozone exposure. Samples were treated with an exposure based on the dosage needed to achieve a reliable virucidal effect. Table 4 shows different types of samples, delivered dose, and the number of total samples treated in the glovebox and trailer system. The respirators and masks were tested intact in both the glovebox and the trailer as discussed in Section 3.2. For mechanical properties testing, the gowns and raw materials were cut to size using a rotary cutter to prevent distortion in pattern lines and fraying, which is important for tensile testing and other mechanical properties testing.

It should be noted that a relatively small sample size of PPEs was used due to the shortage of the products during the COVID19 pandemic. Except for the water impact testing using a single replicate, at least 3–6 replicates were used for other testing. Detailed information about each testing's replicate number was tabulated in supplemental data file (S1 Table). Bar plots with 2σ error bars were generated for each testing. This represents an 80% to 95% confidence

**Table 4. Preparation of PPEs (intact) and material samples for the plasma ROS exposures (glovebox/trailer) and material testing.**

| Sample types | Delivered Dose (ppm-min) | | |
|---|---|---|---|
| | Control | Glovebox | Trailer |
| Proxima Gown | 0(1) | 1800(1), 3700(1) | |
| Polypropylene material | | 700(3), 1200(3), 7000(3) | |
| Polyester material | | 700(3), 1200(3), 7000(3) | |
| 3M N95 (8200) | | 1600(1)*, 1800(1), 3300 (1) | |
| 3M N95 (9502+) | | 1600 (1)* | |
| BYD Respirator | 0 (4)+, | 500(1), 1500(1), 50000 (1)* | 500 (4) +, 1500 (4)+ |
| Prestige Ameritech Respirator | 0 (4)+, | 500(1), 1500(1), 50000 (1)* | 500 (4) +, 1500 (4)+ |
| Prestige Ameritech Gown | 0 (1), | 500(1), 1500(1) | 500(1), 1500(1) |

* Samples inspected at intermediate dose points.

+ Three samples sent to CDC for testing and one for in-house testing.

Number in parenthesis after the dose value is the number of samples treated at that dose.

**Table 5. Experimental parameters used for the Proxima gowns and filter materials in tensile testing.**

| Materials | Specimen Length (mm) | Distance between grips (mm) | Displacement rate (mm/min) |
|---|---|---|---|
| Proxima Gown | 125 | 100 | 300 |
| Polyester | 57.5 | 32.5 | 100 |
| Polypropylene | 57.5 | 32.5 | 100 |
| BYD Respirator | 57.5 | 32.5 | 100 |
| Prestige Ameritech Respirator | 57.5 | 32.5 | 100 |
| Prestige Ameritech Gown | 125 | 100 | 300 |

interval, worse at small sample size. Raw data for individual samples is present in supplemental data file (S2–S22 Tables). In many cases for materials mechanical testing, and respirator performance changes with dose were statistically insignificant. Larger sample size or higher doses may be required to see significant trends.

**3.3.2. Mechanical properties testing.** An Instron 5943 tensile tester with a 1-kN load cell and pneumatic side action grips was utilized for mechanical testing of the PPE materials and straps. These N95 respirators have three layers in which the inner and outer layers are made of polyester and the middle layer is the polypropylene filter, whereas others, such as the BYD respirators, have four layers including a hot air cotton layer. In addition to specimens directly taken from the N95 respirators, raw materials of polyester and polypropylene were tested, as well as Proxima and Prestige Ameritech gown specimens. The testing procedure from ASTM D5035-11 [50] was followed for the polyester and polypropylene filter layers, Proxima gowns, and Prestige Ameritech gowns. ASTM D412-16 [51] and ASTM D638-14 [52] were followed for the polyisoprene straps testing. Table 5 provides details of the specimen dimensions and test speeds. Tensile tests were performed at room temperature (22ºC), and the displacement rate was fixed at 100 mm/min for the materials from the N95 respirators and the polyester and polypropylene raw materials and 300 mm/min for the Proxima and Prestige Ameritech gown specimens, following ASTM D5035.

Fig 8 shows photographs of the ozone treated polypropylene samples before and after tensile testing. The gauge length was drawn across the specimen (horizontal black line) using a fine Sharpie marker. The red lines represent the initial distance between the grips; they aid in

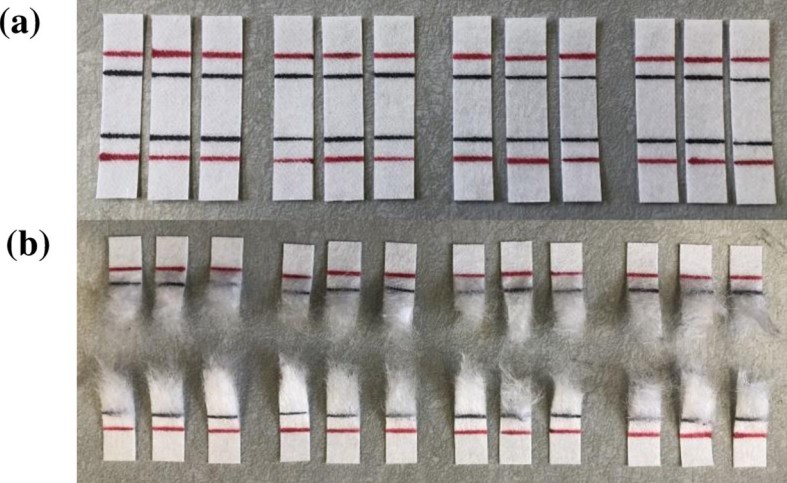

**Fig 8. Ozone treated polypropylene samples (raw material) before (a) and after (b) breaking in the tensile tester.**

proper alignment and in estimating if significant slippage occurred between the specimen and grips during testing.

**3.3.3. Yellowness index testing.** Color change is usually caused by external exposure and will discourage customers to utilize the products even if there is no indication of product decay. Yellowness Index (YI), defined as an indication of the degree of departure of an object color from colorless or from a preferred white toward yellow [53], is used to quantify the extent of color change. This follows standardized test method ASTM E313-15, commonly used to evaluate color changes in a material caused by external exposure. The quantitative evaluation by measuring the YI was done on the N95 respirators before and after treatment. A Nikon D5600 camera was used to take pictures of the ColorChecker palette and sample in the standard light environment (D65). MATLAB software was employed to determine the Yellowness Index at each specifically chosen point at the same specific region on every mask. For the N95 respirators sixty points in each photo were evaluated for their RGB values and yellowness index, two photos of each sample were also analyzed. The region of points in each photo was chosen to be the same general area of the respirator. For smaller articles only ten points in each photo were analyzed. YI for a sample is the mean of the 20 to 120 points (depending on article size), error bars presented in the figures here are 2 times the standard deviation of the multiple points corresponding to about 95% confidence intervals (94% to 95.2%).

**3.3.4. Surface wettability testing.** Surface wettability analysis of the PPE has been carried out to determine changes in the performance of the PPE. Wettability of materials can be characterized by the contact angle, defined as the angle between the liquid-vapor and the solid-liquid interfaces at the point where the three phases (solid, liquid, and gas) meet [54]. Generally, the methods used for contact angle testing have been divided into static drop micro-observation and dynamic testing methods [55]. The static drop micro-observation method was chosen for the quantitative evaluation of wettability, using distilled water droplets resting on the mask material of interest (Fig 9). A Nikon D5600 camera, micro-Nikon lens, and 20 mL syringe were used to image the static drop. The Low-Bond Axisymmetric Drop Shape Analysis (LBADSA) Plugin [56] for ImageJ was employed to determine the contact angle in a given image. For each sample, six repetitions were performed. The average value and 2 times the standard deviation is presented in plots.

**3.3.5. Surface charge measurements.** Filtration efficiency of the respirator material depends not only upon mechanical integrity of the filter material but also on the electrostatic charge, which is applied to the material during manufacturing [57]. Any process of

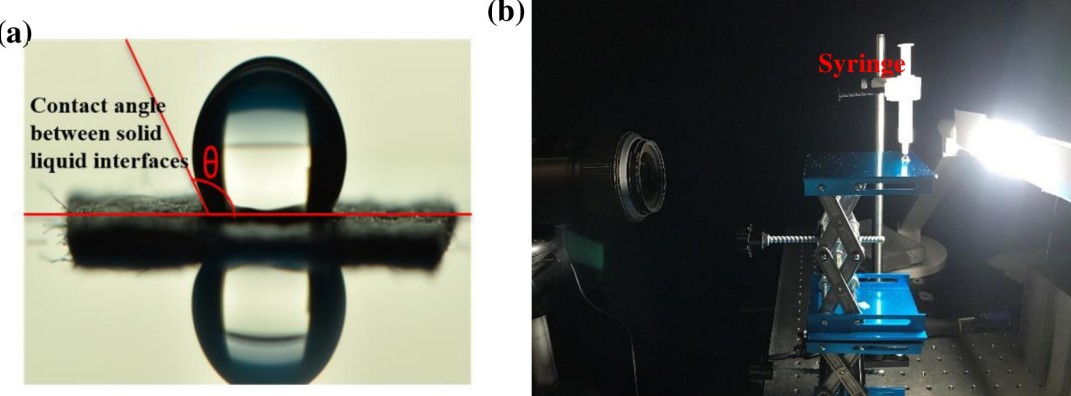

**Fig 9. (a) Contact angle measurement between solid liquid interfaces and (b) surface wettability testing setup.**

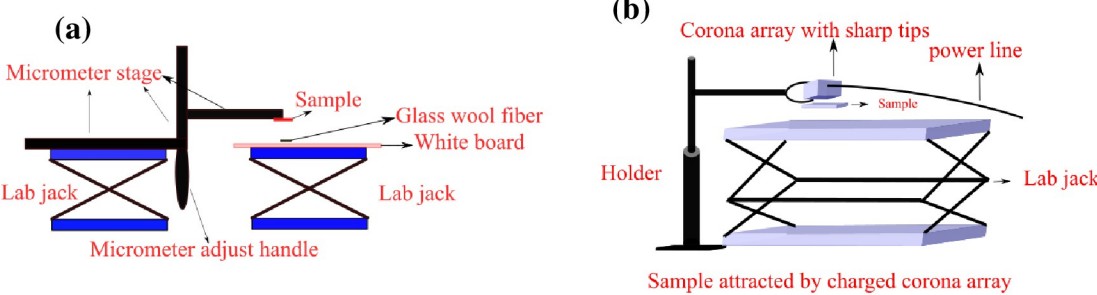

**Fig 10. (a) Surface charge measurement fully assembled setup and (b) corona array for surface charging of the material.**

disinfection may cause loss of the electrostatic charge. Literature suggests that liquids such as saline solutions, distilled water, and alcohols cause to lose the electrostatic charge in the respirator material [58]. The polypropylene materials treated by plasma ROS were tested to measure residual charge function.

A simple setup was prepared to quantify the changes in electrostatic charge on the treated sample via measuring the lift distance of tiny glass wool fibers from various heights to the sample. Fig 10a shows the assembled setup, which comprises a lab jack platform (holding the glass wool at a fixed height), a lab jack holding a micrometer stage, and a whiteboard sitting on the right side of the lab jack. Additionally, a setup was made to intentionally apply an electrostatic charge to the filtration material. Fig 10b shows the corona array setup for charging the sample. A few fibers of glass wool (15 fibers) with 1 cm length were placed on the whiteboard below the sample. The charge measurement was performed before charging, after charging, and on the 15th day after charging to visualize the residual charge and the capability of storing charge of the polypropylene material. The sample is fixed on the micrometer stage just after the charging process and the distance between the sample and the glass wool is gradually decreased until the electrostatic attraction force on the glass wool exceeds gravity and the glass wool "jumps" the remaining distance to the sample. This jumping distance is recorded. This setup is based on the mechanism of electrostatic adsorption. When an object with static electricity is close to another tiny neutral object, due to electrostatic induction, the side of neutral object near the electrostatic object will accumulate a charge of opposite polarity to the charge carried by the charged object and will be adsorbed by the charged object. This adsorption has been well applied on electrostatic adsorption self-assembly process [59, 60], and removal of nitrate [61].

### 3.4. Functionality testing

**3.4.1. Filtration testing.** Both the Prestige Ameritech RP88020 and BYD DE2322 respirators were treated using a plasma ROS method in the trailer at a low ozone dose of 500 ppm-min (1 cycle) and a high ozone dose of 1500 ppm-min (3 cycles). These decontaminated respirators were sent to the National Personal Protective Technology Laboratory, Pittsburgh, for material testing. Three control, three 500 ppm-min, and three 1500 ppm-min exposed respirators were prepared. This low number of samples is due to the shortage of availability from the COVID19 pandemic. NPPTL tested both respirators using a modified version of the NIOSH Standard Test Procedure (STP) TEB-APR-STP-0059 to determine particulate filtration efficiency. The TSI, Inc. model 8130 was used at a flow rate of 85 L/min [62]. The NPPPTL report [62] described the test process: each respirator was tested for 10 minutes, and maximum

penetration was recorded for individual respirator using a sodium chloride aerosol with a maximum concentration of 200 mg/m$^3$.

**3.4.2. Strap testing.** The decontaminated respirators (Prestige Ameritech RP88020 and BYD DE2322) were sent to NPPTL, Pittsburgh for tensile strength testing of the straps. An Instron[R] 5943 tensile tester was used to determine changes in strap integrity. The tensile test was performed by applying the force on bottom and top straps separately. In this test, three control and six decontaminated respirators were used for study. According to the NPPTL report [62], the straps were pulled at 1 cm/s until reaching 150% strain. The samples were then held at 150% strain for 30 seconds, while the force was recorded.

**3.4.3. Exhalation testing.** For exhalation testing, the decontaminated respirators (Prestige Ameritech RP88020 and BYD DE2322) were sent to NPPTL, Pittsburgh. They used a static advanced headform (StAH) to assess the manikin fit factor of respirators [62]. The tube extending from the bottom of the StAH is connected to an inflatable (non-latex, powder-free) bladder inside an isolated and airtight plastic cylinder. This configuration prevents any particles potentially generated by the simulator from entering the breathing zone of the StAH. A port on the cylinder is connected to a Series 1101 breathing simulator (Hans Rudolf, Inc., Shawnee, KS).

**3.4.4. Hydrostatic testing.** A simple hydrostatic pressure tester was developed for the PPE material testing as shown in Fig 11. The PPE was tested using standard AATCC 127 hydrostatic pressure [63]. The setup consisted of 8 feet long PVC tubing and 16 cm-long sanitary tubing. Two PVC valves were used to control the water flow. The pressure-regulating valve was assembled upstream of the setup to monitor the water pressure. The sample was fixed in sanitary tubing with the help of a clamp as shown in Fig 11. The water enters through valve A, only on the closing of valve B. The surface of the gown was observed carefully as the water level continuously rises through the column. As soon as three droplets appear on the

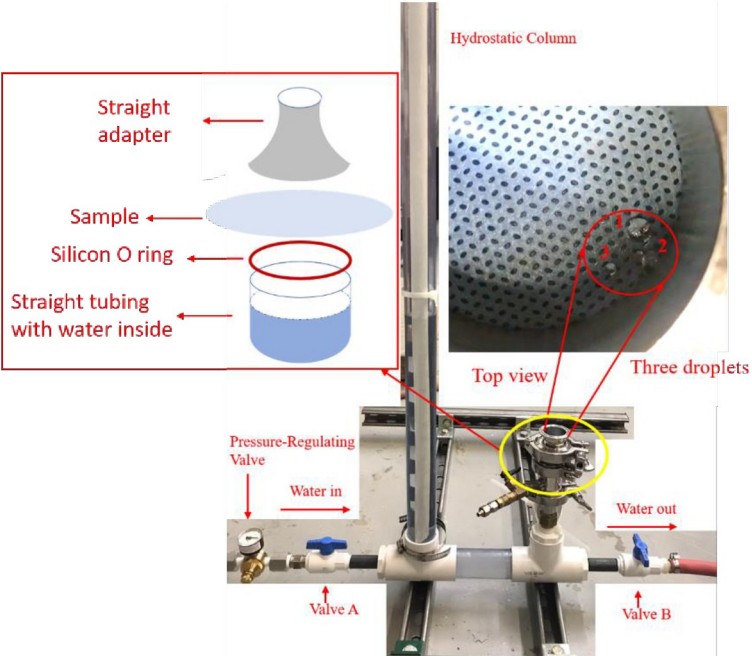

**Fig 11. Hydrostatic pressure tester setup and top view of sample.**

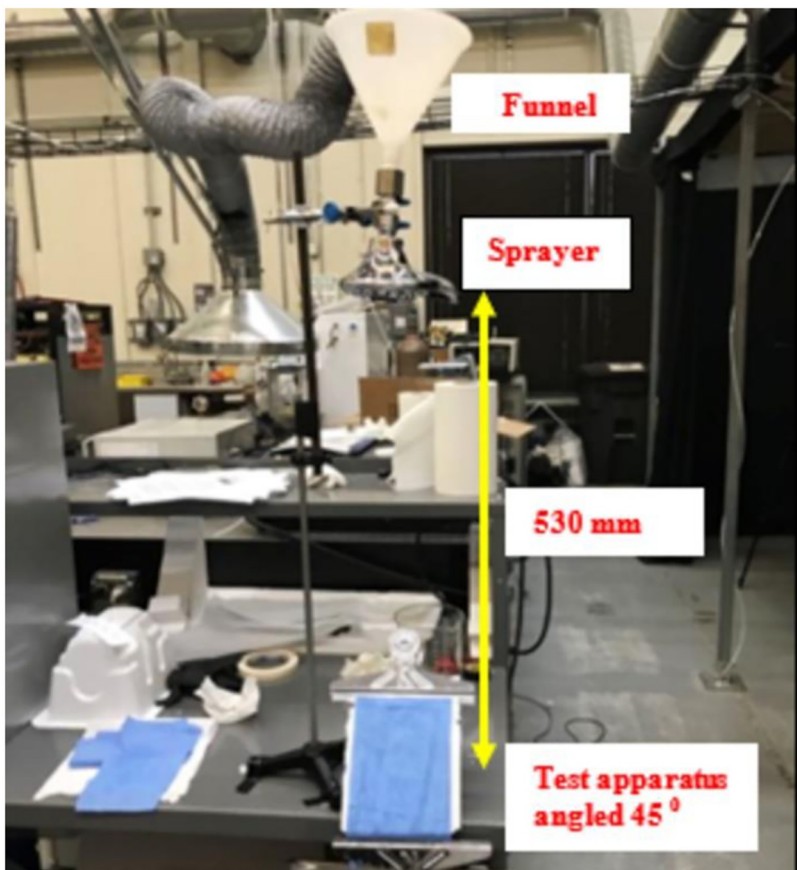

**Fig 12. Schematic of the water impact penetration testing setup.**

surface of the gown, valve A is closed and marked on the PVC tubing. The rise of the water stream is calculated by subtracting the sanitary tubing height from the water column height.

**3.4.5. Water impact penetration testing.** The AATCC 42 based impact penetration standard was used to measure the resistance of fabrics by the impact of water penetration [64]. A 500 mL funnel, a high-pressure showerhead, a 45-degree angle of the test apparatus, and an iron stand were assembled as shown in Fig 12. Gown material samples were prepared with a size of 150 mm and clamped at one end. A smaller size of blotting paper (0.1 gram) was inserted beneath the test sample. A 500 mL volume of distilled water in a 1000 mL beaker was poured into the funnel and allowed to spray onto the test specimen. As the spraying period was accomplished, the test specimen was carefully lifted, and the blotting paper was removed for re-weighing. Finally, the difference of two weights of the blotting paper (before and after the experiment) was observed for analysis.

# 4. Results and discussion

## 4.1. Mechanical testing

Raw materials of polyester/polypropylene (associated with N95 respirators) and the Proxima gown specimens were tested with an Instron tensile tester (performed at TAMU) to determine basic mechanical properties under ozone treatment (PE/PP-Control (0 ppm min), PE/PP-1 (700 ppm min), PE/PP-2 (1200 ppm min), and PE/PP-3 (7000 ppm min)). The Proxima

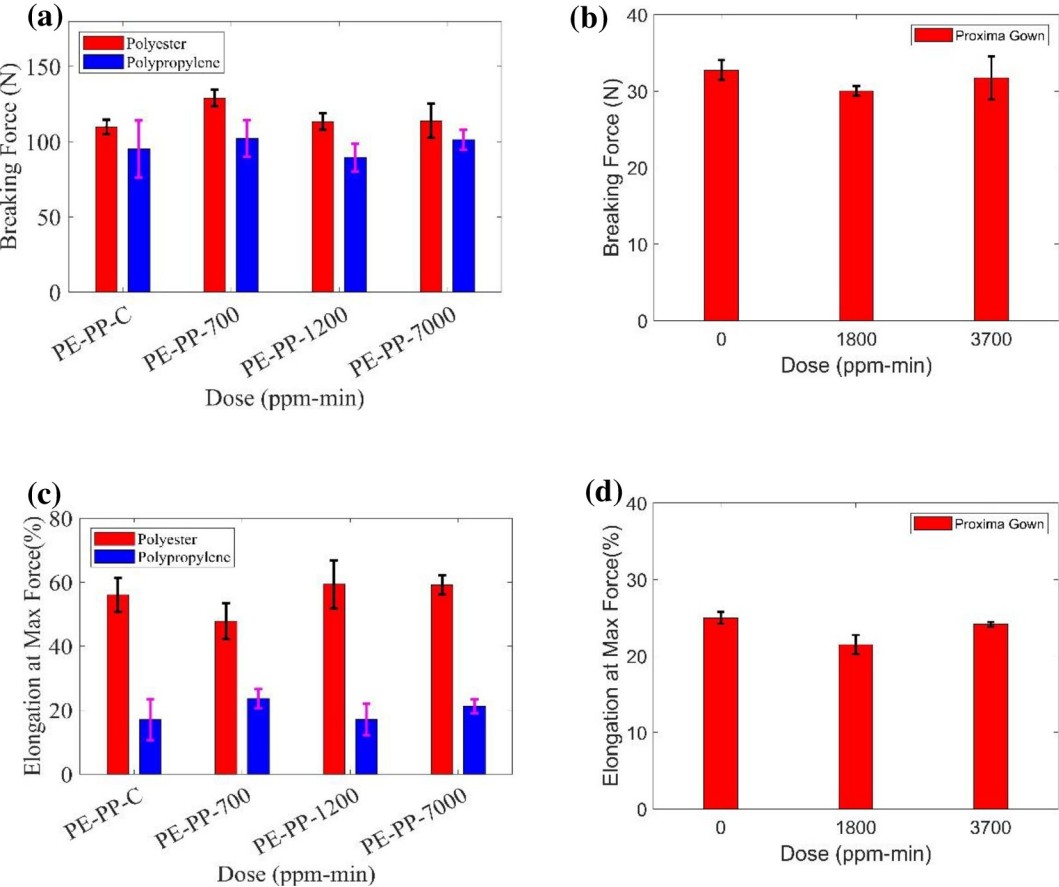

**Fig 13.** Measured breaking force of treated (a) polyester and polypropylene materials and (b) Proxima gown. Measured elongation at max force of treated (c) polyester and polypropylene materials and (d) Proxima gown.

gown materials were treated at an ozone dose of 0 ppm-min (control), 1800 ppm-min, and 3700 ppm-min. ASTM D5035–11 (standard test method for breaking force and elongation of textile fabrics) was generally followed in terms of the testing procedure for the PPE/ materials.

The breaking force of the samples (polyester and polypropylene materials, Proxima gown) from different exposure of ozone doses are shown in Fig 13a and 13b with an error bar representing 2σ from the mean. According to ASTM 5035 and ASTM D4848 [65], the breaking force is defined as the maximum force exerted on the specimen, i.e., the maximum force applied to a material carried to rapture. The elongation at maximum force (%) is determined as a percentage of the length between the grips for the specimen and plotted in Fig 13c and 13d. The breaking force of polyester slightly increases and the elongation at max force slightly decreases for the first dose of 700 ppm-min; however, there is no significant change of the breaking force (N) for polypropylene with ozone dose. For Proxima gowns, there is no significant change of the breaking force with doses of Ozone, while there is a slightly decrease from the control sample to ozone 1800 ppm-min irradiated sample. S2–S5 Tables provide detailed data information.

Additionally, the ozone treated BYD respirators were tested with an Instron tensile tester. Samples were treated at the ozone doses of 500 ppm-min and 1500 ppm-min in the glovebox, as well as 500 ppm-min and 1500 ppm-min in the trailer system. The samples for tensile testing include four nonwoven fabrics of respirator, namely inner (polypropylene spunbond), hot

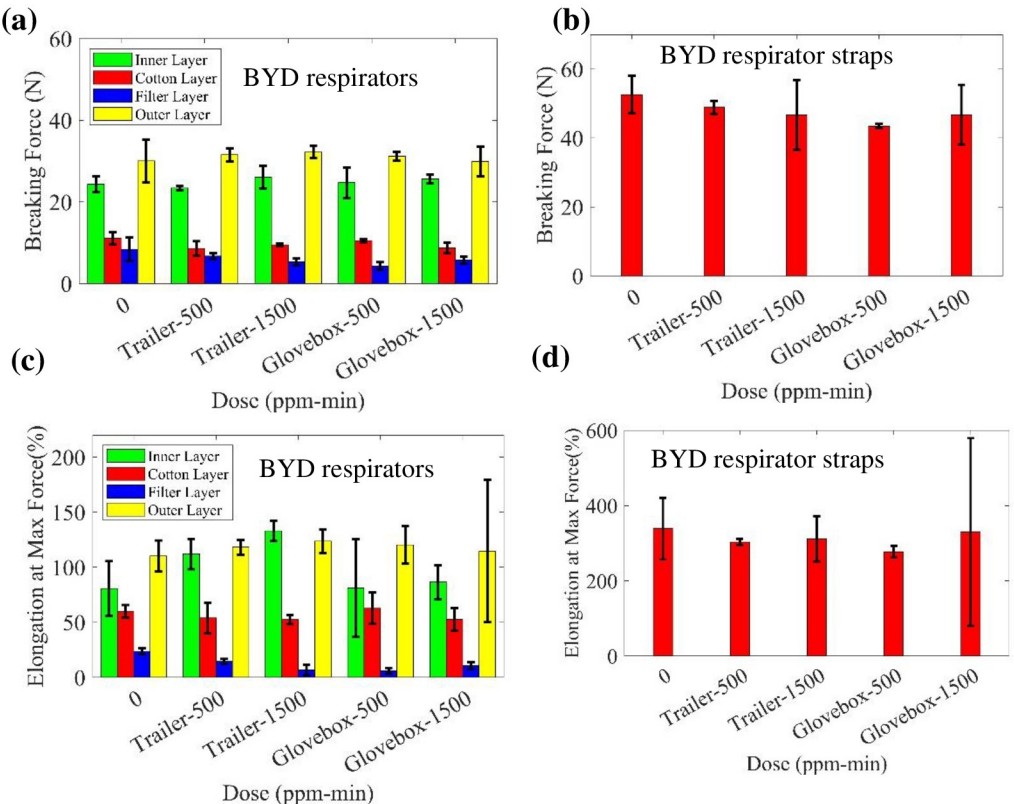

**Fig 14.** Measured breaking force of treated (a) BYD respirators and (b) straps of BYD respirators. Measured elongation at max force of treated (c) BYD respirators and (d) straps of BYD respirators.

air cotton, filter (polypropylene melt-blown), and outer (polypropylene spunbond) layers, as well as the strap material.

The breaking force of the samples from different doses are shown in Fig 14a and 14b. The error bars represent two standard deviations. The elongation at maximum force (%) for the specimens is presented in Fig 14c and 14d. There is no significant negative change of the breaking force and elongation at maximum force for the three layers of BYD respirators with ozone dose; however, for the filter layer, the properties decrease slightly with dose. Considering the difference between glovebox and trailer, the difference of 500 ppm-min treated BYD filter layer's elongation at max force between trailer and glovebox treatment was obviously noticed, which is unexpected, and it may result from the limited tested sample size of 3.

Similarly, the ozone-treated Prestige Ameritech (PA) respirators, straps, and gowns were tested with an Instron tensile tester. Samples were treated at the ozone dose of 500 ppm-min and 1500 ppm-min in the glovebox, as well as 500 ppm-min and 1500 ppm-min in the trailer system. The sample for tensile testing includes three layers of fabric (inner, middle and outer) and strap material. The breaking force of the samples from different doses are shown in Fig 15a–15c, and the elongation at maximum force of the PA respirators and gown is shown in Fig 15d–15f. The breaking force of the PA gown treated in glovebox decreases slightly with dose relative to the control; otherwise, no significant changes with dose were observed. No obvious difference between samples treated in glovebox and trailer appears in the measurements of breaking force and elongation at max force.

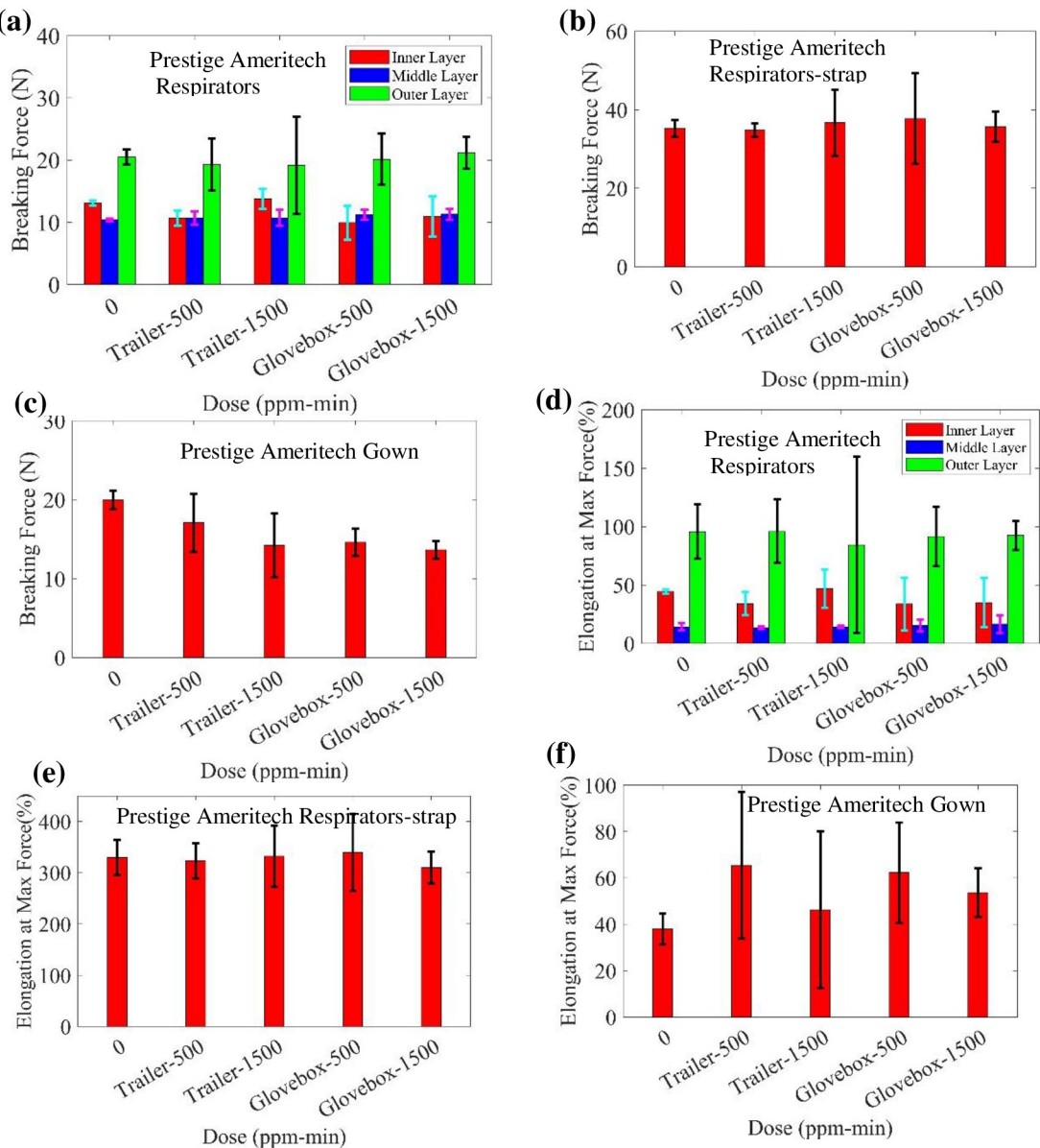

**Fig 15.** Measured breaking force of treated (a) Prestige Ameritech respirators, (b) straps of respirators, and (c) gowns. Measured elongation at max force of treated (d) Prestige Ameritech respirators, I straps of respirators, and (f) gowns.

## 4.2. Yellowness index testing

The color change of plasma ROS treated samples was evaluated by giving the value of yellowness index (YI), a quantitative number based on X, Y, Z color space. A change in YI of about 5 is noticeable to humans. The inside and outside of N95 respirators, front and back sides of polypropylene and polyester materials, and both sides of BYD and Prestige Ameritech respirators have been processed to analyze the YI. Fig 16a represents the difference value of YI between control sample and treated sample, indicating no significant difference occurs in the polypropylene (PP) and polyester (PE) materials. The large error bars are due to low uniformity and large sample areas used for analysis. More importantly, the yellowness index differences (ΔYI)

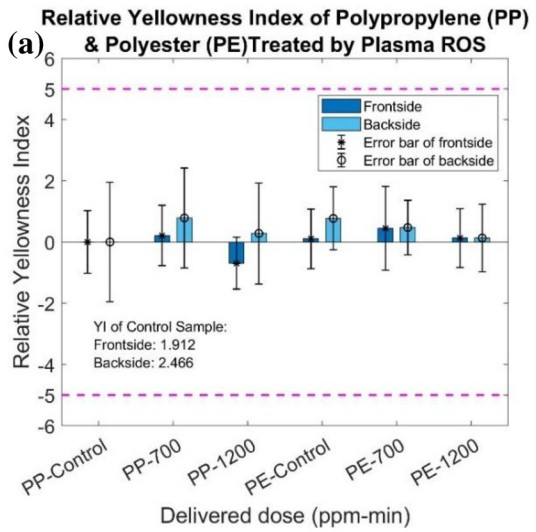
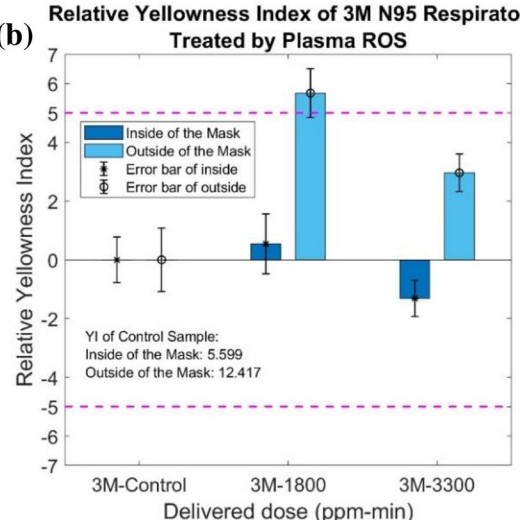

**Fig 16. Yellowness index of (a) polypropylene and polyester material and (b) 3M N95 respirators decontaminated by plasma ROS.**

between the control sample and treated samples are all below 5, which is effectively imperceptible to the human eyes. The yellowness index of the 3M N95 respirator, as seen in Fig 16b, shows little dependence on the delivered dose. It is observed that the outside of the 3M respirator has a higher YI compared to the inside surface. Meanwhile, the ΔYI of 1800 ppm-min treated sample is slightly bigger than 5, which may result from the limited test samples.

The bar chart of Fig 17 presents the relative YI values (relative to the control sample) of BYD, Prestige Ameritech respirators, straps, and gowns. There are no visually observable differences in YI in terms of the different delivered doses (control, 500 ppm-min and 1500 ppmmin) and different treated systems. Also, all ΔYI of treated sample are within or close to 5.

## 4.3. Hydrostatic pressure testing

A self-assembled hydrostatic pressure tester was equipped to measure the hydrostatic pressure of Proxima and Prestige Ameritech gowns treated by plasma ROS. Table 6 shows the values of the hydrostatic pressure of Proxima and Prestige Ameritech gowns, which were treated at two different ozone doses. Note that the maximum pressure able to be recorded with this system is 3.194 psi, which was exceeded by many samples. A moderate water resistant gown should have hydrostatic pressure higher than 0.71 psi according to the standard reported by CDC [66]. Only the Proxima gown with a delivered dose of 3722 ppm-min shows poor results of the testing, being below the 0.71 psi pressure threshold. All trailer treated samples (both 1 and 3 cycles of the Prestige Ameritech Gown) pass this test.

## 4.4. Surface wettability testing

Wettability of the PPE materials treated by plasma ROS was estimated by contact angle measurements. The experiments for each material were repeated six times for the testing. Results of water contact angle of Proxima, as well as polyester and polypropylene materials, were boxplotted shown in Fig 18a–18c. Fig (a) shows that the frontside of Proxima Gown has negligible differences with the control average value. However, the backside of the Proxima Gown indicates a large decrease of the Proxima-3700 compared with Proxima-Control and Proxima-1800. The results of both polypropylene (PP) and polyester (PE) indicate a continuous

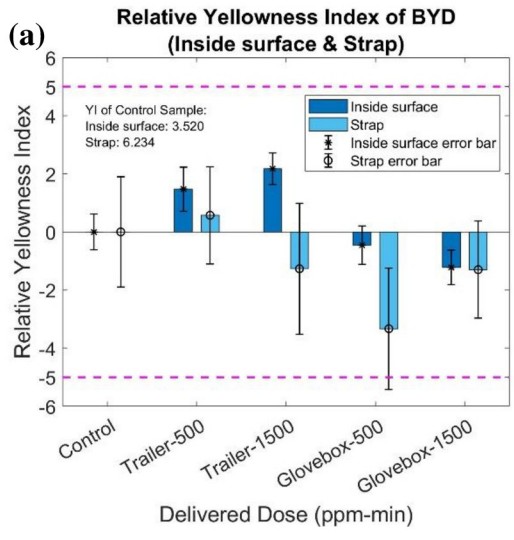
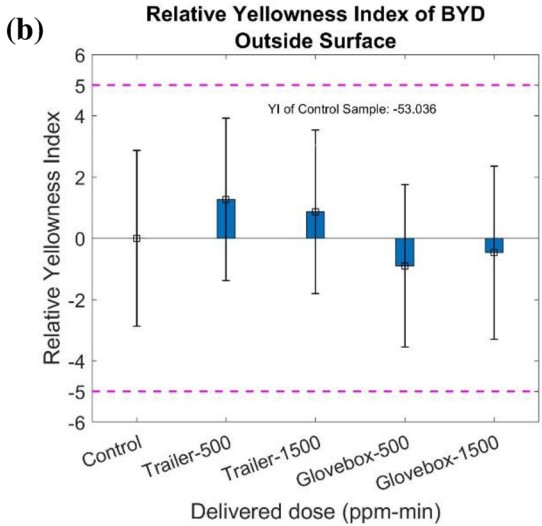
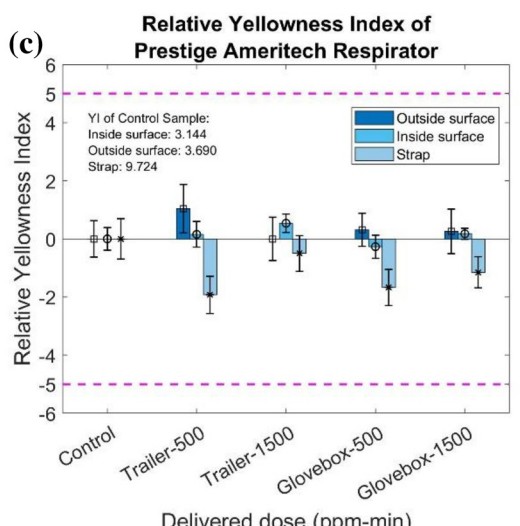
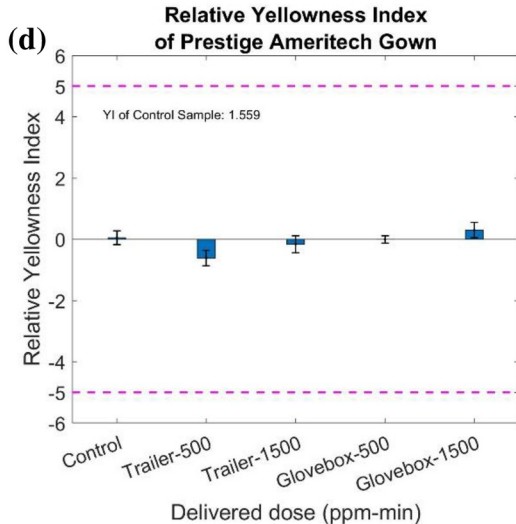

**Fig 17. Yellowness index of (a) the BYD respirator inside surface and strap, (b) BYD respirator outside surface, (c) Prestige Ameritech respirators and strap, and (d) Prestige Ameritech gown treated by plasma ROS.**

**Table 6. Hydrostatic pressure results of Proxima and Prestige Ameritech gowns treated by PlasmaROS.**

| Sample | Proxima-C | Proxima-1800 ppm- min | Proxima-3700 ppm- min | PA-C | PA-Trailer-500 ppm min | PA-Glovebox-500 ppm min | PA-Trailer-1500 ppm min | PA-Glovebox-1500 ppm min |
|---|---|---|---|---|---|---|---|---|
| Repetition 1 | 1.191 | 1.265 | 0.451 | >3.194 | 3.086 | >3.194 | 3.194 | >3.194 |
| Repetition 2 | 1.267 | 1.164 | 0.684 | >3.194 | 3.194 | >3.194 | >3.194 | 3.045 |
| Repetition 3 | 1.182 | 1.135 | 0.721 | >3.194 | >3.194 | >3.194 | >3.194 | 2.901 |
| Average | 1.213 | 1.188 | 0.618 | >3.194 | >3.16 | >3.158 | >3.194 | >3.047 |
| STDEV | 0.038 | 0.056 | 0.119 | N/A | N/A | N/A | N/A | N/A |

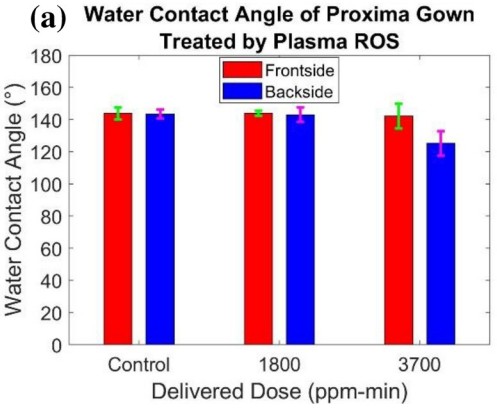
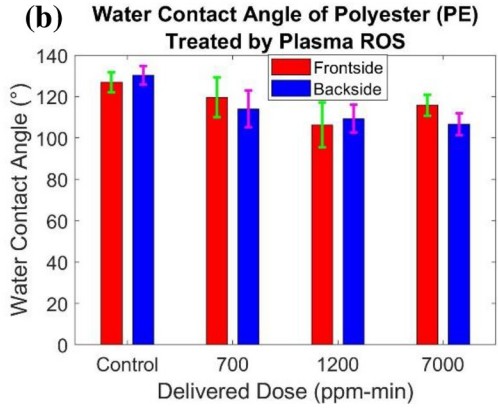

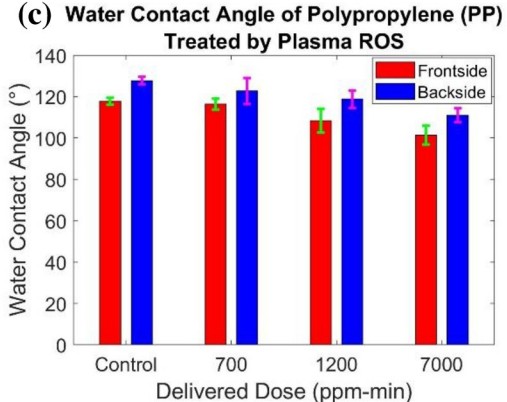

**Fig 18. Water contact angle of (a) Proxima gown, (b) polyester, and (c) polypropylene samples treated by Plasma ROS method.**

decrease of water contact angle with an increase of delivered dose occurring both at the frontside and backside. However, the situation of the frontside of PE-7000 does not follow this rule. In summary, plasma ROS treatment introduces relatively negative effects on the hydrophobic property, but they still maintain general hydrophobicity with a contact angle greater than 90°.

## 4.5. Impact penetration testing

A simple method was implemented for impact penetration testing based on the AATCC test method 42–2017. According to this standard, the testing guidelines of moderate water resistance gowns should be satisfied with the Proxima and PA gowns. It is noted from the standard that the weight gain of blotting paper should be less than 1 gram [66]. For the limited samples treated, the Proxima and PA gowns passed the requirement. Fig 19 shows the variation of weight gain of blotting papers for differently treated Proxima and PA gowns. All the samples that performed impact penetration testing successfully passed the testing, independent of manufacture or dose. There are slight increases at the very high dose for the Proxima, but still less than 9% of the threshold to meet the standard.

## 4.6. Surface charge measurement testing

Surface charge was measured for polypropylene (PP) control samples and treated samples exposed to 700 ppm-min, 1200 ppm-min, and 7000 ppm-min dose levels. As discussed

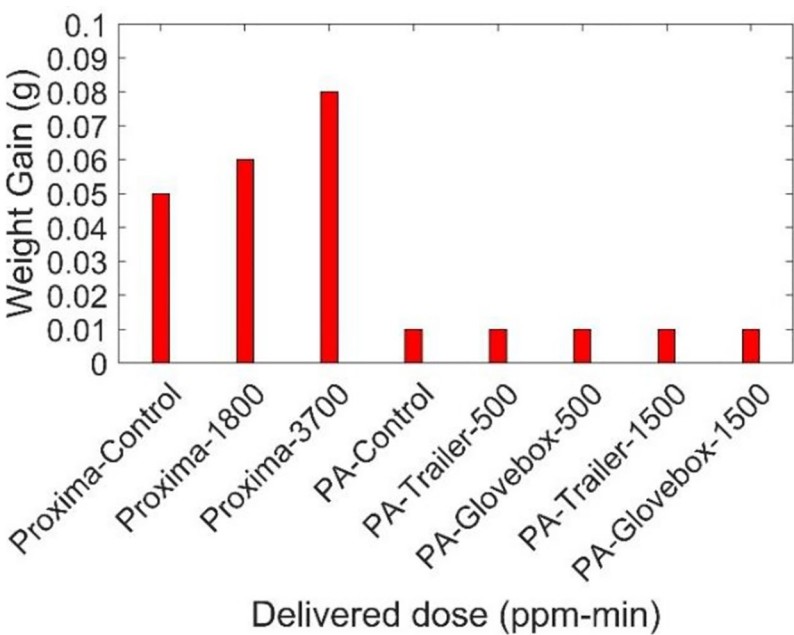

**Fig 19. Impact pentration testing results of different Proxima and Prestige Ameritech gown samples.**

previously (in Section 3.3.5), the corona array setup is used to quantify the changes in electrostatic charge on the treated sample in terms of the 'lift distance' parameter. Fig 20 shows the lift distance of polypropylene material from before charging, immediately after charging, and 15 days after charging. An average value from six repetitive experiments was employed for surface charge analysis.

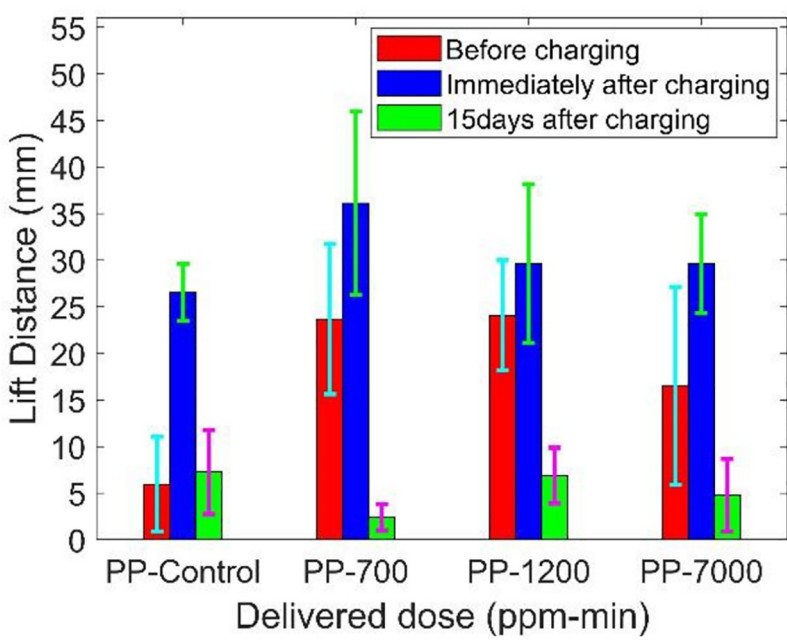

**Fig 20. Surface charge measurement of polypropylene material treated by Plasma ROS.**

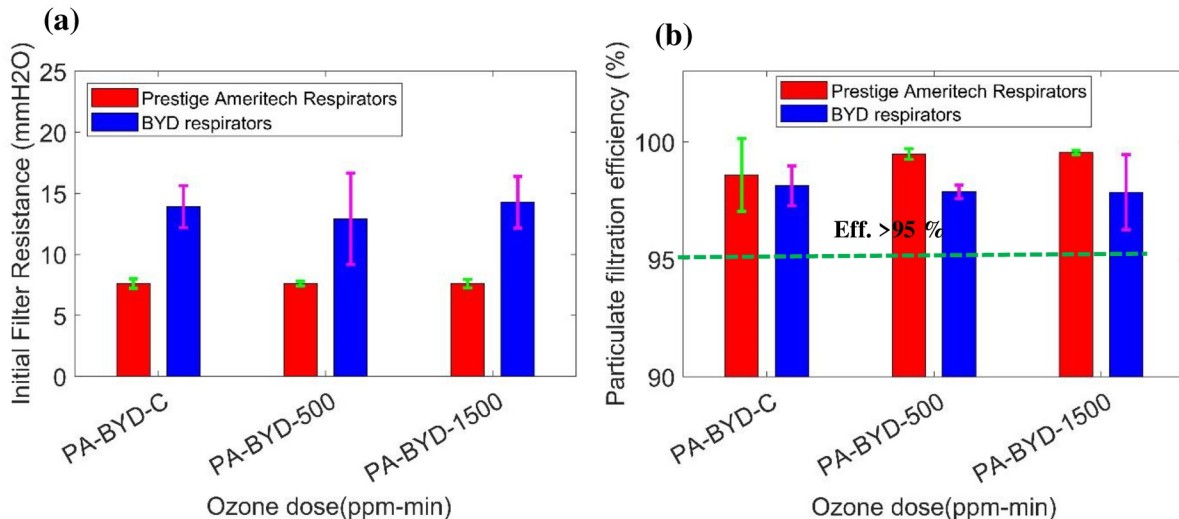

**Fig 21. (a)** Initial filter resistance and **(b)** particulate filtration efficiency of Prestige Ameritech and BYD respirators (control samples and 1 (500 ppm-min) and 3 (1500 ppm-min) decontaminated cycle of plasma ROS).

It is observed from the Fig that the lift distance measured from plasma ROS treated PP samples show relatively higher values compared with the control samples since plasma ROS treatment put charges on the samples. Ion concentrations in Aerisa 5350 generated ozone have been tested in a duct with a flow rate of 620 cfm (0.3 m3/s). Results, as seen in Table 3, indicate the number of ion particles is generated along with ozone and these particles contribute to the higher residual surface charge present on the plasma ROS treated samples. Both positive and negative ions are generated, their concentration is on the order of several part per trillion. This is significantly lower than the ozone concentration. The lift distances measured immediately after charging and measured 15 days after charging indicate that the plasma ROS treatment did not significantly change the ability of the material to hold charges. The charges applied during the plasma ROS treatment of respirators did not alter the respirator filter efficiency as discussed in Section 4.7.

## 4.7. Particulate filtration testing

An assessment was developed to quantify the filtration efficiency and manikin fit factor of N95 respirators (Prestige Ameritech RP88020 and BYD DE2322) by the NPPTL group. A number of three control of each type respirators (PA-C, BYD-C), 500 ppm-min exposed samples (PA-1cycle, BYD-1 cycle), and 1500 ppm-min exposed samples (PA-3 cycle, BYD-3cycle) were utilized for the analysis. Fig 21a shows a bar chart of the initial filter resistance of respirators based on treatment dose values. The error bars represent two standard deviations. It is observed from the graph that there is no significant difference in initial filter resistance due to plasma ROS treatment. In addition, the values of initial filter resistance of the BYD type respirator are higher compared to Prestige Ameritech's resistance values. Similarly, Fig 21b shows particulate filter efficiencies for the control samples (PA-C, BYD-C), samples treated by 500 ppm-min exposure (PA-1cycle, BYD-1 cycle), and samples treated by 1500 ppm-min exposure (PA-3 cycle, BYD-3cycle). It is observed from the graph that there is no significant variation in filtration efficiency due to an increment in ozone dose level. Ranges of filter efficiency of 99.35–99.56%, 99.50–99.59%, 97.79–98.04% and 97.10–98.69% were observed for the PA-1cycle, PA-3 cycle, BYD-1 cycle, and BYD-3cycle samples, respectively. The overall

particulate filter efficiencies of all treated respirators exhibit greater than 95% efficiency. There is no significant difference observed in each condition group.

## 4.8. Tensile testing

Strap integrity testing (Instron 5943 tensile tester) was performed by the NPPTL group. Tensile force in the top and bottom straps of respirators (treated by plasma ROS) was recorded at 150% strain. Fig 22a shows values of tensile force for the control respirators (PA-C, BYD-C), 500 ppm-min exposed samples (PA-1cycle, BYD-1 cycle), and 1500 ppm-min exposed samples (PA-3 cycle, BYD-3cycle). There is not much difference in tensile force due to plasma ROS exposure in the top straps of the respirators. Similarly, Fig 22b shows almost equal tensile force observed in the bottom straps of the control and treated samples. The CDC reported no visual degradation of the straps after the plasma ROS exposure and BYD respirators have better performance compared with Prestige Ameritech Respirators.

An in-house test was performed to evaluate the integrity of straps of different types of respirators. Two respirators—a 3M 9502+ and a 3M 8200 –were subjected to ozone exposure of 1600 ppm-min at an ozone concentration of approximately 20 ppm. During treatment of the 3M 8200 respirator, the straps of the respirator broke off at around 1000 ppm-min. They started wearing off at around 400 ppm-min (Fig 23a). The 9502+ respirator was intact after treatment to 1600 ppm-min ozone exposure as shown in Fig 23b. However, the 3M 8200 respirator failed at the location of metallic staples where the straps are attached to respirator (see Fig 23c and 23d).

To understand the failure behavior of 3M 8200 respirator straps, polyisoprene (strap material of 3M 8200 respirators) samples were exposed to ozone inside the glove box at different dose levels. In the first setup the straps were arranged in the glovebox flat without introducing any physical stresses (Fig 24a). In a second setup (Fig 24b) straps were induced to bend over the support causing a stress at one point. In addition, a piece of copper tape was placed along the strap to inspect whether charge deposition on the metallic staples is causing the failure at that particular location in the 3M 8200 respirator. After each cycle of ozonation process, the straps were cyclically stretched to twice their initial length ten times each to determine whether their mechanical properties had changed during treatment. In a third setup pre-stretched

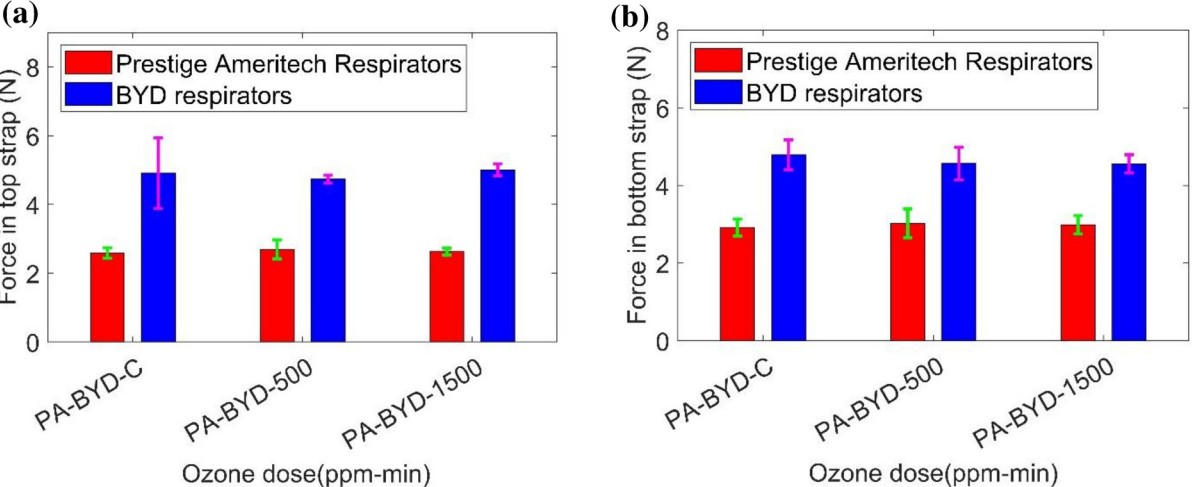

**Fig 22. Tensile force in the respirators (Prestige Ameritech and BYD) (a) top strap and (b) bottom strap.**

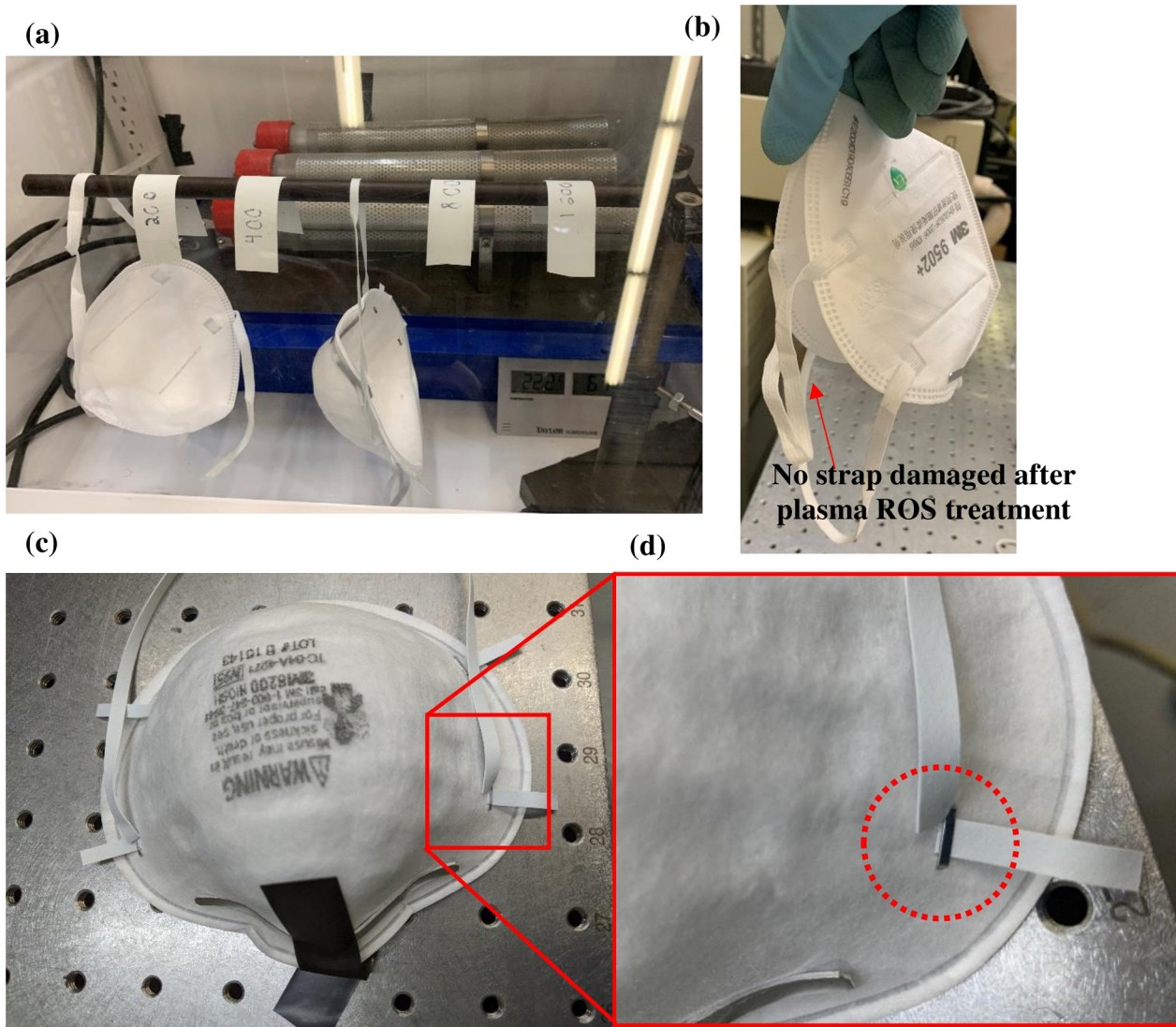

**Fig 23.** (a) Respirators during glovebox testing after treatment to 1600 ppm-min ozone exposure. (b) 3M 9502+ and (c) 3M 8200. (d) A zoomed-in view of the damaged 3M 8200 respirator straps after treated at 1600 ppm-min ozone dose.

samples (stretched to double their length) were placed inside the glovebox treated by ozone (Fig 24d).

After ozone exposure, the setup without induced physical stress did not show any physical damage even after 1600 ppm-min. However, straps from the setup with the specimens bent over the support started showing physical damage around 400 ppm-min near the bend region. No damage was observed near the copper tape. At around 1000 ppm-min the wear at the bend propagated throughout the width, and the straps broke into two pieces. In the final setup with stretched straps, physical damage was first observed at the point where there was a twist on the strap material. These results indicate that it is the concentrated physical stress that leads to damage. In the case of the 3M 8200 respirator, concentrated stress is induced in strap material by the metal staple, which ultimately leads to damage.

Such behavior of strain and stress accelerated degradation of polymers has been observed before using similar experimental configurations. The degradation of polymer straps induced

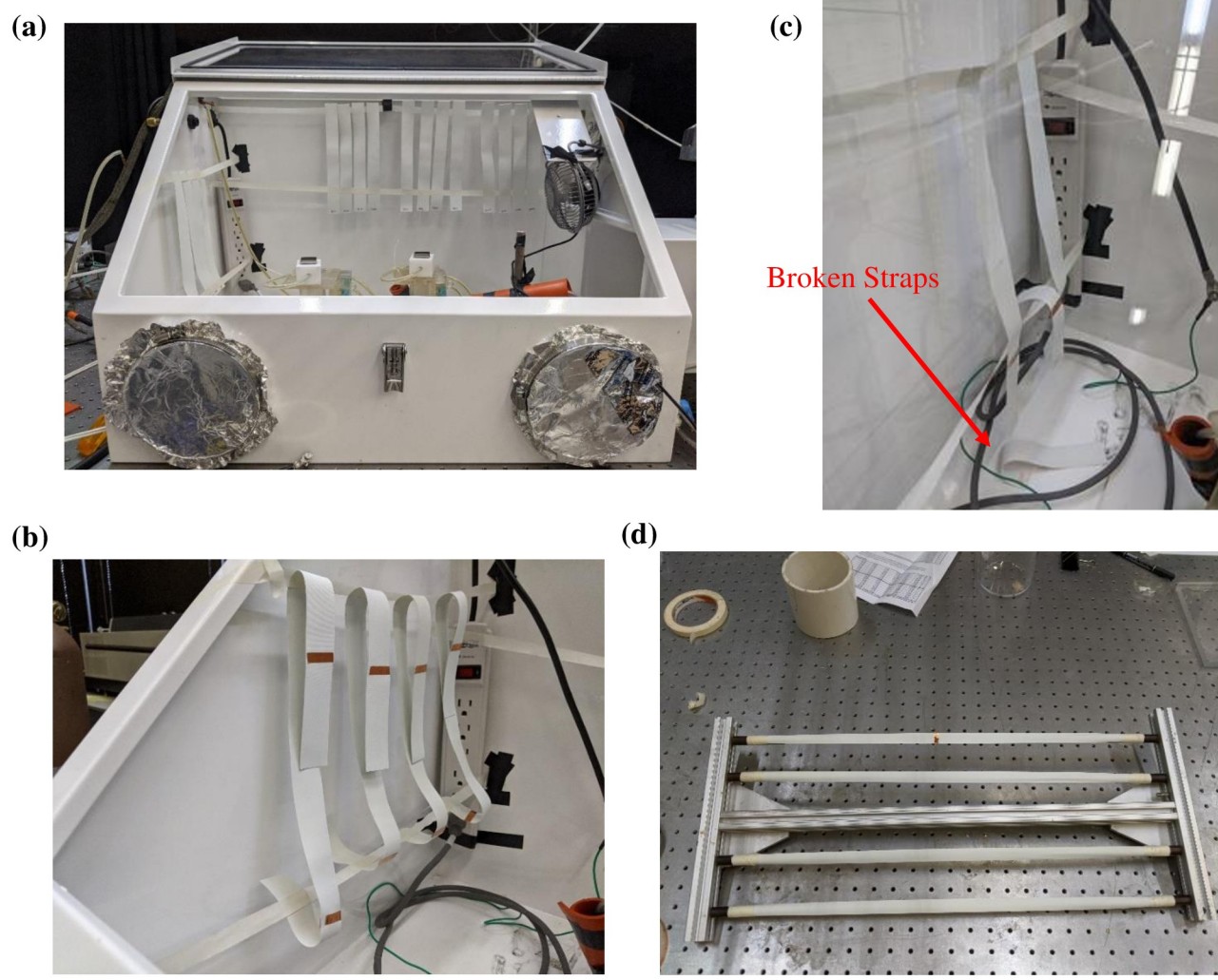

**Fig 24. Strap (polyisoprene) arrangement in glovebox testing: (a) straps placed without any stress, (b) straps placed with a stress, (c) broken strap after treated by 1000 ppm-min dose, and (d) straps elongated to double their initial lengths.**

by ozone oxidation discussed above can be explained by oxidative crosslinking and oxidative aging, which lead to decreased elastic modulus of elastomers and stress cracking [67–70]. The relationship between applied strain on polymer sheets and the growth of surface crack with exposure in ozone was studied by loading a parallel-sided strip of the rubber with ordinary weights in Andrews et al's work [71] and by placing in tension jigs in the ozone chamber in Kamaruddin's work [72]. Higher strain resulting in higher number of surface cracks in both ozone concentration of 1ppm (1mg/L) and 0.5 ppm (50 pphm) in above references was observed. In a study by Beachell et al. in both polystyrene and polyethylene, where most of the active sites on the surface have been destroyed, diffusion of oxidizing gas in to the polymer becomes the rate controlling process [73]. These reference materials used similar testing methods to ours and help to explain and confirm our observations.

The strap degradation was only observed in 3M strap materials (Polyisopylene). For both Prestige Ameritech and BYD respirators even at very high ozone dose of up to 50000 ppm-min (the equivalent of about 100 cycles) no failure of straps is observed. The straps of both

respirators (polyester/nylon spandex and combination of Lycra and polyester materials) were cyclically stretched to twice their initial length ten times. However, no physical damage was observed after plasma ROS treatment.

## 5. Conclusion

A mobile trailer was repurposed to facilitate plasma ROS treatment for decontamination of PPEs to deal with emergency reuse during the COVID-19 pandemic. Several types of N95 respirators (Prestige Ameritech, BYD, and 3M) and gowns (Prestige Ameritech and Proxima) were exposed to low dose levels of 500 ppm-min (1-cycle) and high dose levels of 1500 ppm-min (3-cycles) in a closed system. Data from literature shows that 500 ppm-min ozone dose (in the gaseous form) is capable of achieving $\geq$ 3-log reduction in non-enveloped virus or vegetative bacteria. Exposure of longer period of ozone dose (1500 ppm-min) should be capable of fully decontaminating the PPE. According to the above Table 2 listing line powers, discharge powers, and corresponding electrical efficiency of Aerisa DBD generators, maximum power consumed for the trailer will be ~180 W. With 35-m$^3$ volume in the trailer, there can be 57 stainless racks installed for placing N95 respirators. ~24 respirators sitting per rack leading to a total of 1380 respirators can be treated per cycle. To optimize exposure to ozone, respirators were placed separately such that they did not touch each other. The system is overall easy to use. The results provided by'CDC's NPPTL and Texas A&M demonstrate the effectiveness of this mobile trailer system for reducing the bioburden of PPE for reuse. The reliability of the Aerisa DBD generators allow for minimal upkeep required on the system.

Although the high ozone concentration causes potential safety issues, the trailer system is well sealed when operating to minimize any potential ozone exposure to users. Ozone is exhausted and levels measured below 100 ppb prior to human entry. In addition, considering the safety risk caused by residual partial oxidation products and ozone on the PPEs, appropriate aeration is needed. During treatment of PPEs, ozone directly reacts with organic compounds of the targeted viruses and PPEs. The rapid decomposition and vaporization of ozone in appropriate aeration also diminish the residue risk. Although ozone and hydrogen peroxide, which are the possible harmful residues on PPE are detectable based on previous research, proper post-decontamination aeration was also indicated to be helpful with the elimination of residues [74]. New compounds generated during ozone decomposition may also be residuals. The potential skin contact risk caused by residual hydrogen peroxide can also be reduced by increasing aeration time, air exchange rate, and temperature.

Therefore, the mobile trailer presented in this paper is a promising system for healthcare sectors and industries leading to bioburden reduction or decontamination of PPE during emergency reuse. Our technique has been published on the CDC website as report 5 and report 27 [75].

In this paper, decontaminated PPEs (including N95 respirators, gowns, and filter materials) were evaluated by assessing mechanical integrity and materials performance after exposure to different ozone dose levels. Furthermore, the PPE decontamination processes were repeated with small-scale (glovebox) testing and larger-scale (trailer) testing for verification. PPE treated with 1-cycle and 3-cycles has been tested by CDC NPPL. Based on these results operation cycle should be always limited to 3 cycles considering the uncertainty of dysfunctional PPE treated with high cycles. However, future research works could be done to verify PPE functionality at higher number of cycles.

The ASTM specific material testing such as yellowness index testing, surface wettability testing, and surface charge measurement testing were performed at TAMU for the PPE/

material samples. For the post-processing of samples, material functionality testing was performed for PPE which includes particulate filtration testing, strap integrity testing, exhalation testing, hydrostatic testing, and water impact penetration testing. The particulate filtration, respirator fit on a manikin head, and strap integrity tests were performed at NPPTL, Pittsburgh. There is no visible degradation observed in the material and strap testing with increasing ozone dose levels for BYD and Prestige Ameritech respirators. Results also demonstrate that straps (3M 8200 respirator) made from polyisoprene failed but straps made from polyester/nylon spandex and combination of Lycra and polyester materials (BYD and Prestige Ameritech) passed.

Filter efficiencies were found to be greater than 95% for the samples (Prestige Ameritech and BYD respirators) treated by ozone doses between 0–1500 ppm-min. We show that filtration efficiency does not depend on surface charge variation in plasma ROS-treated material samples. The respirators (BYD, Prestige Ameritech) retained their filtration performance and tensile strength by decontamination in the larger-scale mobile trailer testing. These results suggest that plasma ROS and ozone treatment is a potential candidate for bioburden reduction or decontamination of PPE to meet the demand during the COVID-19 pandemic. It is worth noting that this testing only takes into account damage to new PPE caused by the decontamination process. There may be other typical use factors which would additionally damage used PPE. A PPE decontamination and reuse protocol would first require inspection of the used PPE to ensure that field use did not damage the PPE. Only PPE which is intact and functional should be subject to a decontamination process. Considering both field use and decontamination induced damage the practical application of PPE recycling is reduced. Testing and decontamination of used PPE was beyond the scope of this project. Cross correlated effects or combined field use damage and decontamination damage is possible but unknown at this point.

## Supporting information

**S1 Table. Replicates number for each material characterization method.**
(DOCX)

**S2 Table. Internal tensile testing for polypropylene.**
(DOCX)

**S3 Table. Internal tensile testing for Proxima gown.**
(DOCX)

**S4 Table. Internal tensile testing for BYD three layers and band.**
(DOCX)

**S5 Table. Internal tensile testing for BYD three layers and band.**
(DOCX)

**S6 Table. Internal tensile testing for Prestige Ameritech gown.**
(DOCX)

**S7 Table. Internal tensile testing for Prestige Ameritech mask.**
(DOCX)

**S8 Table. Yellowness index testing for polypropylene.**
(DOCX)

**S9 Table. Yellowness index testing for polyester.**
(DOCX)

**S10 Table. Yellowness index testing for 3M N95 respirator.**
(DOCX)

**S11 Table. Yellowness index testing for BYD DE2322.**
(DOCX)

**S12 Table. Yellowness index testing for Prestige Ameritech respirator.**
(DOCX)

**S13 Table. Yellowness index testing for Prestige Ameritech gown.**
(DOCX)

**S14 Table. Surface wettability testing for Proxima gown.**
(DOCX)

**S15 Table. Surface wettability testing for Proxima gown.**
(DOCX)

**S16 Table. Surface wettability testing for polypropylene.**
(DOCX)

**S17 Table. Water impact penetration testing for Proxima gown and Prestige Ameritech gown.**
(DOCX)

**S18 Table. Surface charge measurement for polypropylene.**
(DOCX)

**S19 Table. Particulate filtration testing for Prestige Ameritech respirator.**
(DOCX)

**S20 Table. Particulate filtration testing for BYD DE2322.**
(DOCX)

**S21 Table. Strap tensile testing for Prestige Ameritech respirator.**
(DOCX)

**S22 Table. Strap tensile testing for BYD DE2322.**
(DOCX)

**S1 Graphical abstract.**
(TIF)

## Acknowledgments

The PPE testing (filtration efficiency and strap integrity) was performed by NIOSH NPPTL in Pittsburgh, Pennsylvania.

## Author Contributions

**Conceptualization:** Min Huang, Md Kamrul Hasan, Kavita Rathore, Md Abdullah Hil Baky, John Lassalle, Jamie Kraus, Matthew Burnette, Howard Jemison, Suresh Pillai, Matt Pharr, David Staack.

**Data curation:** Min Huang, Md Abdullah Hil Baky, John Lassalle, David Staack.

**Formal analysis:** Min Huang, Md Kamrul Hasan, Kavita Rathore, Md Abdullah Hil Baky, Matthew Burnette, David Staack.

**Investigation:** Min Huang, Md Kamrul Hasan, Kavita Rathore, Md Abdullah Hil Baky, John Lassalle, Jamie Kraus, Matthew Burnette, Christopher Campbell.

**Methodology:** Min Huang, Md Kamrul Hasan, Kavita Rathore, Md Abdullah Hil Baky, John Lassalle, Jamie Kraus, Matthew Burnette, Kunpeng Wang, Howard Jemison, Suresh Pillai, Matt Pharr, David Staack.

**Project administration:** Kavita Rathore, David Staack.

**Resources:** Suresh Pillai, Matt Pharr, David Staack.

**Supervision:** Suresh Pillai, Matt Pharr, David Staack.

**Writing – original draft:** Min Huang, Kavita Rathore, Md Abdullah Hil Baky, John Lassalle, Matthew Burnette, David Staack.

**Writing – review & editing:** Min Huang, Md Kamrul Hasan, Kavita Rathore, Md Abdullah Hil Baky, John Lassalle, Jamie Kraus, Matthew Burnette, Christopher Campbell, Kunpeng Wang, Howard Jemison, Suresh Pillai, Matt Pharr, David Staack.

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
