## [Decision Letter · Decision Letter 0]

16 Nov 2021

PONE-D-21-23319

Plasma Generated Ozone and Reactive Oxygen Species for Point of Use PPE Decontamination System

PLOS ONE

Dear Dr. Huang,

Thank you for submitting your manuscript to PLOS ONE. After careful consideration, we feel that it has merit but does not fully meet PLOS ONE’s publication criteria as it currently stands. Therefore, we invite you to submit a revised version of the manuscript that addresses the points raised during the review process.

We look forward to receiving your revised manuscript.

Kind regards,

Amitava Mukherjee, ME, Ph.D.

Academic Editor

PLOS ONE

https://journals.plos.org/plosone/s/file?id=wjVg/PLOSOne_formatting_sample_main_body.pdf and https://journals.plos.org/plosone/s/file?id=ba62/PLOSOne_formatting_sample_title_authors_affiliations.pdf2.

Reviewers' comments:

Reviewer's Responses to Questions

**Comments to the Author**

1. Is the manuscript technically sound, and do the data support the conclusions?

Reviewer #1: Partly

2. Has the statistical analysis been performed appropriately and rigorously? 

Reviewer #1: No

3. Have the authors made all data underlying the findings in their manuscript fully available?

Reviewer #1: Yes

4. Is the manuscript presented in an intelligible fashion and written in standard English?

Reviewer #1: Yes

5. Review Comments to the Author

Reviewer #1: This paper reports the DBD plasma reactor-based ozone and reactive oxygen species generation for the decontamination of respirators and gowns. This timely work mainly tested the functionality/stability of the N95 respirators and gowns after 1 and 3 cycles of decontamination in the glovebox and mobile trailer system. However, most of the experiments were conducted with only one sample, this could undermine the validity of the study.

Due to fewer sample size, there was no significant correlation on effect magnitudes of decontamination cycles against different properties tested. Please, provide raw experimental data as supplement information.

The observed structural damage in the brand new PPEs during 3 cycles of decontamination underscores that there may more damage in the used PPEs. Hence, the efficacy of the used PPEs after the first decontamination cycle needs to be tested to validate as a commercially viable technique and the number of recycles that could be possibly achieved in practical remains naïve.

Since this study was designed to disinfect the PPEs using ozone and reuse during this COVID-19 pandemic, the possible dose and mechanism of disinfection of SARS-CoV-2 should be discussed in the review or conclusion section.

As most of the studies, this study also reported only the physical property of the tested material, it is essential to validate remaining gaseous ozone residues and derivatives on the PPEs because these residues/ derivatives on the PPEs may pose inhalation and/or dermal exposure risk.

6. PLOS authors have the option to publish the peer review history of their article (what does this mean?). If published, this will include your full peer review and any attached files.

Reviewer #1: **Yes: **Gopinath PM

---

## [Author Response · Author response to Decision Letter 0]

31 Dec 2021

Responses to reviewer’s comments: 

Note: Yellow-highlighted is the reviewer’s comments we specifically addressed, and responses are in maroon font color

Reviewer #1: This paper reports the DBD plasma reactor-based ozone and reactive oxygen species generation for the decontamination of respirators and gowns. This timely work mainly tested the functionality/stability of the N95 respirators and gowns after 1 and 3 cycles of decontamination in the glovebox and mobile trailer system. However, most of the experiments were conducted with only one sample, this could undermine the validity of the study.

(Yes, only a limited number of samples was implemented due to the shortage of the products during the COVID19 pandemic. The number of samples treated is listed in table 4. However, we would like to point out that for each respirator sample sent for external lab testing, 3 replicates were used. For internal material testing a single treated sample was cut into multiple pieces for replication of the material and functionality testing. A paragraph has been added to section 3.3.1 to clarify this.)

Due to fewer sample size, there was no significant correlation on effect magnitudes of decontamination cycles against different properties tested. Please, provide raw experimental data as supplement information.

(Raw tabulated data on results from individual trials has been provided as supplemental information. 

On all plots 2𝜎 is shown, this has been explicitly stated as a general statement about statistics in the new paragraph in section 3.3.1. In some cases, only a single replicate test was performed.

You are correct that for most cases the effect of ozone dose does not cause statistically significant property changes. i.e. we cannot reject the null hypothesis. However, it is not clear that a larger sample size would necessarily lead to a significant correlation with dose (at least over the dose range tested) in all cases. At higher doses, we do expect degradation. )

The observed structural damage in the brand-new PPEs during 3 cycles of decontamination underscores that there may more damage in the used PPEs. Hence, the efficacy of the used PPEs after the first decontamination cycle needs to be tested to validate as a commercially viable technique and the number of recycles that could be possibly achieved in practical remains naïve.

(You are correct that this testing only takes into account damage to new PPE caused by the decontamination process. There may be other typical use factors that would additionally damage used PPE. A PPE decontamination and reuse protocol would first require inspection of the used PPE to ensure that field use did not damage the PPE. Only PPE which is intact and functional should be subject to a decontamination process. Taking into account both field use and decontamination-induced damage the practical application of PPE recycling is reduced. Testing and decontamination of used PPE were beyond the scope of this project. Cross correlated effects or combined field use damage and decontamination damage are possible but unknown at this time. This stipulation has been added to the ‘conclusion’ section.)

Since this study was designed to disinfect the PPEs using ozone and reuse during this COVID-19 pandemic, the possible dose and mechanism of disinfection of SARS-CoV-2 should be discussed in the review or conclusion section.

(The possible dose for bioburden reduction and decontamination has been discussed in the literature review section. This does not particularly address SAR-CoV-2 as literature on that is not available yet. However, the applicable mechanism of disinfection is translatable to other viruses like SARS-CoV-2 and additional explanation of this has been added to the ‘literature review’ section, and more detailed information has been discussed in the ‘reaction mechanism’ section: 

While SAR-CoV-2 (or any coronavirus) is not specifically addressed by this literature review viruses, similar enveloped viruses presented in Table 1 is Φ6. Vaccinia virus, vesicular stomatitis virus, yellow fever virus, and Sendai virus are all enveloped viruses, and potential explanations for their ozone inactivation are protein capsid damage and genome degradation. There is a great agreement on the alterations, which are induced by ozone in the lipids and proteins present in the membrane of these viruses [1]. As a preferred method used for the generation of ozone, corona discharge also provides another potential mechanism for the decontamination of viruses. Although reactive oxygen and/or nitrogen species (RONS) play the most important role in sterilizing microorganisms, ultraviolet radiation induces direct damage to microorganisms and bacteria via breakage of DNA/RNA and chemical alterations of bases due to the absorption of high energy. More detailed information is covered in section 3.1.2 regarding direct ozone-induced and indirect ozone-generated oxidant species-induced damages. Increasingly attention on ozone inactivation of the enveloped virus is aware, and the mechanism has been discussed in recent researches [2–4]. Research also indicates that induced changes in the viral RNA genome, appearing as aggregation and fusion of influenza viruses (also known as enveloped viruses) were observed via SEM after N2 gas plasma treatment [5].)

As most of the studies, this study also reported only the physical property of the tested material, it is essential to validate remaining gaseous ozone residues and derivatives on the PPEs because these residues/ derivatives on the PPEs may pose inhalation and/or dermal exposure risk.

(The reviewer is likely referring 1) to residual partial oxidation products and oxidized PPE plastics which may have odor or toxicity and 2) residual ozone on the PPE. Regarding 2: Less than 100 ppb ozone level after treatment in glovebox and trailer systems was required to achieve prior to human touch. Also, the rapid decomposition and vaporization of ozone in appropriate aeration diminish the residue risk. Regarding 1: During the treatment of PPEs, ozone will directly react with organic compounds of the targeted viruses and PPE materials. These new compounds may also be residuals. Although ozone and hydrogen peroxide, which are the possible harmful residues on PPE are detectable based on previous research, proper post-decontamination aeration was also indicated to be helpful with the elimination of residues [6]. The potential skin contact risk caused by residual hydrogen peroxide can also be diminished by increasing aeration time, air exchange rate, and temperature.) A discussion of these potential issues has been added to the ‘conclusion’ section. )

Reference

1. Bayarri B, Cruz-Alcalde A, López-Vinent N, Micó MM, Sans C. Can ozone inactivate SARS-CoV-2? A review of mechanisms and performance on viruses. J Hazard Mater. 2021;415. 

2. Dubuis ME, Racine É, Vyskocil JM, Turgeon N, Tremblay C, Mukawera E, et al. Ozone inactivation of airborne influenza and lack of resistance of respiratory syncytial virus to aerosolization and sampling processes. Vol. 16, PLoS ONE. 2021. 

3. Grignani E, Mansi A, Cabella R, Castellano P, Tirabasso A, Sisto R, et al. Safe and Effective Use of Ozone as Air and Surface Disinfectant in the Conjuncture of Covid-19. Gases. 2020;1(1):19–32. 

4. Blanchard EL, Lawrence JD, Noble JA, Xu M, Joo T, Ng NL, et al. Enveloped Virus Inactivation on Personal Protective Equipment by Exposure to Ozone. medRxiv : the preprint server for health sciences. 2020. 

5. Sakudo A, Shimizu N, Imanishi Y, Ikuta K. N2 gas plasma inactivates influenza virus by inducing changes in viral surface morphology, protein, and genomic RNA. Biomed Res Int. 2013;2013. 

6. Kumkrong P, Scoles L, Brunet Y, Baker S, Mercier PHJ, Poirier D. Evaluation of hydrogen peroxide and ozone residue levels on N95 masks following chemical decontamination. J Hosp Infect [Internet]. 2021;111:117–24. Available from: https://doi.org/10.1016/j.jhin.2021.02.018

---

## [Decision Letter · Decision Letter 1]

6 Jan 2022

Plasma Generated Ozone and Reactive Oxygen Species for Point of Use PPE Decontamination System

PONE-D-21-23319R1

Dear Dr. Huang,

We’re pleased to inform you that your manuscript has been judged scientifically suitable for publication and will be formally accepted for publication once it meets all outstanding technical requirements.

Kind regards,

Amitava Mukherjee, ME, Ph.D.

Academic Editor

PLOS ONE

Additional Editor Comments (optional):

Reviewers' comments:

Reviewer's Responses to Questions

**Comments to the Author**

1. If the authors have adequately addressed your comments raised in a previous round of review and you feel that this manuscript is now acceptable for publication, you may indicate that here to bypass the “Comments to the Author” section, enter your conflict of interest statement in the “Confidential to Editor” section, and submit your "Accept" recommendation.

Reviewer #1: (No Response)

2. Is the manuscript technically sound, and do the data support the conclusions?

Reviewer #1: (No Response)

3. Has the statistical analysis been performed appropriately and rigorously? 

Reviewer #1: (No Response)

4. Have the authors made all data underlying the findings in their manuscript fully available?

Reviewer #1: (No Response)

5. Is the manuscript presented in an intelligible fashion and written in standard English?

Reviewer #1: (No Response)

6. Review Comments to the Author

Reviewer #1: (No Response)

7. PLOS authors have the option to publish the peer review history of their article (what does this mean?). If published, this will include your full peer review and any attached files.

Reviewer #1: **Yes: **Ponnusamy Manogaran Gopinath

---

## [Editor Report · Acceptance letter]

10 Feb 2022

PONE-D-21-23319R1 

Plasma Generated Ozone and Reactive Oxygen Species for Point of Use PPE Decontamination System 

Dear Dr. Huang:

I'm pleased to inform you that your manuscript has been deemed suitable for publication in PLOS ONE. Congratulations! Your manuscript is now with our production department. 

Kind regards, 

on behalf of

Professor Dr. Amitava Mukherjee 

Academic Editor

PLOS ONE